# An ER translocon for multi-pass membrane protein biogenesis

**Philip T McGilvray[1,2†], S Andrei Anghel[1,2†], Arunkumar Sundaram[1], Frank Zhong[1,2], Michael J Trnka[3], James R Fuller[1], Hong Hu[4], Alma L Burlingame[3], Robert J Keenan[1]***

[1]Department of Biochemistry and Molecular Biology, The University of Chicago, Chicago, United States; [2]Department of Molecular Genetics and Cell Biology, The University of Chicago, Chicago, United States; [3]Department of Pharmaceutical Chemistry, University of California, San Francisco, San Francisco, United States; [4]Center for Research Informatics, The University of Chicago, Chicago, United States

**Abstract** Membrane proteins with multiple transmembrane domains play critical roles in cell physiology, but little is known about the machinery coordinating their biogenesis at the endoplasmic reticulum. Here we describe a ~ 360 kDa ribosome-associated complex comprising the core Sec61 channel and five accessory factors: TMCO1, CCDC47 and the Nicalin-TMEM147-NOMO complex. Cryo-electron microscopy reveals a large assembly at the ribosome exit tunnel organized around a central membrane cavity. Similar to protein-conducting channels that facilitate movement of transmembrane segments, cytosolic and luminal funnels in TMCO1 and TMEM147, respectively, suggest routes into the central membrane cavity. High-throughput mRNA sequencing shows selective translocon engagement with hundreds of different multi-pass membrane proteins. Consistent with a role in multi-pass membrane protein biogenesis, cells lacking different accessory components show reduced levels of one such client, the glutamate transporter EAAT1. These results identify a new human translocon and provide a molecular framework for understanding its role in multi-pass membrane protein biogenesis.

*For correspondence:
bkeenan@uchicago.edu

†These authors contributed equally to this work

Competing interests: The authors declare that no competing interests exist.

## Introduction

The human genome encodes thousands of integral membrane proteins, which play critical roles in nearly all aspects of cell physiology. Membrane proteins of the cell surface and most intracellular compartments are first assembled at the endoplasmic reticulum. Most of these are inserted by the evolutionarily conserved Sec61 complex, which guides hydrophobic transmembrane domains (TMDs) into a central aqueous channel that opens laterally to allow TMD entry into the bilayer (*Voorhees and Hegde, 2016*; *Li et al., 2016*; *Pfeffer et al., 2015*).

How this process is elaborated to facilitate the insertion and folding of membrane proteins containing multiple TMDs is not well understood. The human genome encodes ~2500 multi-pass proteins, including GPCRs, solute carriers, ion channels, and ABC transporters. These show considerable biophysical and topological complexity, including TMDs of variable length and hydrophobicity, closely spaced TMD hairpins, and re-entrant loops that span only part of the membrane (*Cymer et al., 2015*; *Foster et al., 2000*). These features are often critical for function, but they pose a significant challenge for the biosynthetic machinery (*Foster et al., 2000*; *Tector and Hartl, 1999*).

The 'translocon' is a poorly defined and dynamic ensemble that coordinates the insertion, folding, modification and assembly of most membrane proteins. The eukaryotic translocon comprises the core Sec61 channel in association with different accessory factors. The best studied of these factors include the OST (*Chavan et al., 2005*) and TRAP complexes (*Fons et al., 2003*), TRAM (*Görlich and*

**eLife digest** Cell membranes are structures that separate the interior of the cell from its environment and determine the cell's shape and the structure of its internal compartments. Nearly 25% of human genes encode transmembrane proteins that span the entire membrane from one side to the other, helping the membrane perform its roles.

Transmembrane proteins are synthesized by ribosomes – protein-making machines – that are on the surface of a cell compartment called the endoplasmic reticulum. As the new protein is made by the ribosome, it enters the endoplasmic reticulum membrane where it folds into the correct shape. This process is best understood for proteins that span the membrane once. Despite decades of work, however, much less is known about how multi-pass proteins that span the membrane multiple times are made.

A study from 2017 showed that a protein called TMCO1 is related to a group of proteins involved in making membrane proteins. TMCO1 has been linked to glaucoma, and mutations in it cause cerebrofaciothoracic dysplasia, a human disease characterized by severe intellectual disability, distinctive facial features, and bone abnormalities. McGilvray, Anghel et al. – including several of the researchers involved in the 2017 study – wanted to determine what TMCO1 does in the cell and begin to understand its role in human disease.

McGilvray, Anghel et al. discovered that TMCO1, together with other proteins, is part of a new 'translocon' – a group of proteins that transports proteins into the endoplasmic reticulum membrane. Using a combination of biochemical, genetic and structural techniques, McGilvray, Anghel et al. showed that the translocon interacts with ribosomes that are synthesizing multi-pass proteins. The experiments revealed that the translocon is required for the production of a multi-pass protein called EAAT1, and it provides multiple ways for proteins to be inserted into and folded within the membrane.

The findings of McGilvray, Anghel et al. reveal a previously unknown cellular machinery which may be involved in the production of hundreds of human multi-pass proteins. This work provides a framework for understanding how these proteins are correctly made in the membrane. Additionally, it suggests that human diseases caused by mutations in TMCO1 result from a defect in the production of multi-pass membrane proteins.

*Rapoport, 1993*; *Voigt et al., 1996*), Sec62/63 (*Conti et al., 2015*), and the signal peptidase complex (*Kalies et al., 1998*). A significant challenge to studying the translocon is that many accessory factors only transiently or sub-stoichiometrically associate with the core machinery, contributing to difficulties in isolating intact complexes (*Görlich and Rapoport, 1993*; *Wang and Dobberstein, 1999*). As a result, the composition and stoichiometry of ribosome-bound translocons, and their structures, functions and clientele, remain poorly defined.

We previously identified TMCO1 as a eukaryotic member of the Oxa1 superfamily, whose members are linked to membrane protein biogenesis (*Anghel et al., 2017*). These proteins, including the EMC3 subunit of the 'ER membrane complex' (EMC) (*Chitwood et al., 2018*; *Guna et al., 2018*; *Shurtleff et al., 2018*; *Volkmar et al., 2019*), the Get1 subunit of the Get1/2 complex (*Schuldiner et al., 2008*; *Mariappan et al., 2011*; *Wang et al., 2014*), and the Oxa1/Alb3/YidC proteins (*Shanmugam and Dalbey, 2019*), function in different contexts as TMD insertases and/or as intramembrane chaperones to facilitate membrane protein folding and assembly (*Shurtleff et al., 2018*; *Nagamori et al., 2004*; *Serdiuk et al., 2016*; *Klenner et al., 2008*). The function of TMCO1 is not yet known, but consistent with a role in a co-translational process at the ER membrane, it can be natively isolated in association with ribosome-Sec61 complexes (*Anghel et al., 2017*).

## Results

### Interaction partners of natively isolated TMCO1-ribosome complexes

To identify components of TMCO1-ribosome complexes we solubilized microsomes isolated from 3xFlag-TMCO1 HEK293 cells, affinity purified via the Flag tag on TMCO1, isolated the ribosome-bound fraction by sedimentation, and identified co-purifying proteins by quantitative mass

spectrometry (*Figure 1A,B*). Ribosomal proteins, subunits of the Sec61 complex, and TMCO1 were enriched relative to control cells lacking the Flag tag, whereas known translocon accessory factors—including subunits of the OST and TRAP complexes, TRAM, Sec62/63 and the signal peptidase complex—were either weakly enriched or absent. We also observed strong enrichment of three poorly studied proteins: the single-pass membrane protein CCDC47 (calumin) and two subunits of the Nicalin-TMEM147-NOMO transmembrane complex (*Dettmer et al., 2010*).

We confirmed recovery of Sec61, TMCO1, Nicalin, TMEM147, NOMO and CCDC47 in the ribosome-associated fraction following 3xFlag-TMCO1 immunoprecipitation (*Figure 1C,D*). Notably, the catalytic OST subunit STT3A was not detected, consistent with the absence of OST from the TMCO1-ribosome complexes. Affinity purification via a Flag tag on Nicalin recovered TMCO1, CCDC47 and NOMO in the ribosome-bound fraction (*Figure 1—figure supplement 1A*), indicating that these proteins can be isolated as a single, ribosome-associated complex. In the absence of ribosomes, however, only components of the Nicalin-TMEM147-NOMO complex remained intact (*Figure 1—figure supplement 1B*), suggesting that TMCO1, CCDC47 and the pre-formed Nicalin-TMEM147-NOMO complex assemble in the context of the ribosome.

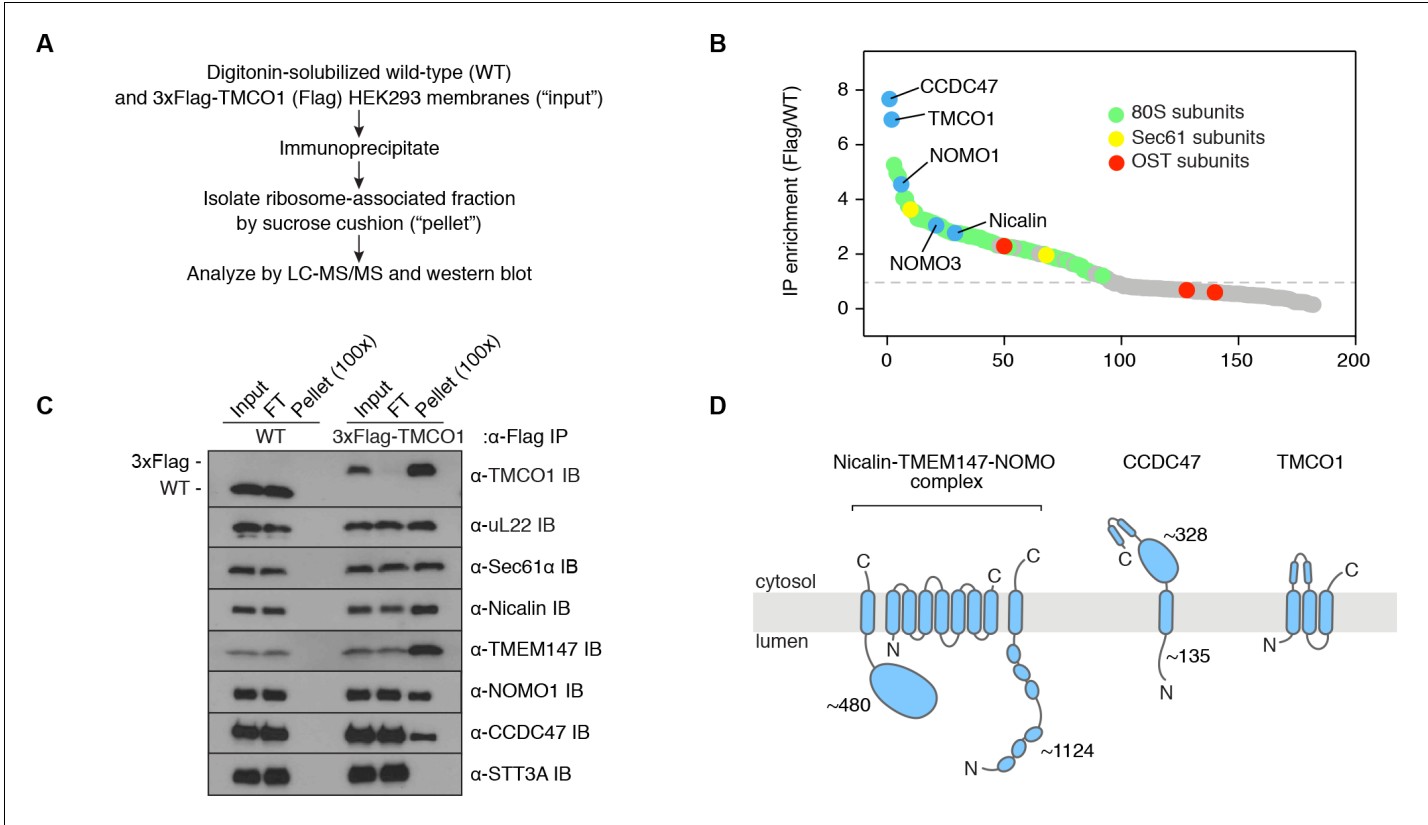

**Figure 1.** Natively isolated TMCO1-ribosome complexes contain multiple transmembrane components. (A) Emetine- and micrococcal nuclease-treated membranes from wild-type (WT) or 3xFlag-TMCO1 (Flag) HEK293 cells were digitonin-solubilized, immunoprecipitated via the 3xFlag tag on TMCO1, and the eluate sedimented through a sucrose cushion to isolate the ribosome-associated fraction for analysis. (B) Proteins enriched in the ribosomal fraction after immunoprecipitation from 3xFlag-TMCO1 or wild-type membranes. (C) Top hits were confirmed by western blotting. The catalytic STT3A subunit of the OST complex is not detected. (D) Topology and domain structure for the top hits, based on Uniprot annotation, homology modeling, de novo structure prediction (in RaptorX-Contact), and experimental mapping; the Sec61 complex is not shown. Distinguishing features include the large globular luminal domain of Nicalin (in contrast with the flexible luminal domains of NOMO and CCDC47), the large globular cytosolic domain of CCDC47 (with a conserved C-terminal coiled-coil), and a conserved cytosolic coiled-coil between the first two TMDs of TMCO1. TMEM147 is the core, multi-pass subunit of the Nicalin-TMEM147-NOMO complex *Dettmer et al., 2010*; note that the short extra-membrane loops of TMEM147 make it difficult to detect by mass spectrometry.

The online version of this article includes the following figure supplement(s) for figure 1:

**Figure supplement 1.** Additional interaction analysis of the TMCO1 translocon components.

CCDC47, Nicalin, TMEM147 and NOMO are abundant ER-localized proteins, conserved across eukaryotes, widely expressed in human tissues, and associated with several human diseases (*Itzhak et al., 2016*; *Burdon et al., 2011*; *Sharma et al., 2012*; *Xin et al., 2010*; *Caglayan et al., 2013*; *Alanay et al., 2014*; *Li et al., 2019*; *Reuter et al., 2017*; *Morimoto et al., 2018*). Although their functions remain obscure, CCDC47 has been linked to various membrane-associated processes (*Morimoto et al., 2018*; *Zhang et al., 2007*; *Konno et al., 2012*; *Thapa et al., 2018*; *Yamamoto et al., 2014*), and the Nicalin-TMEM147-NOMO complex has been proposed to regulate subunit assembly and localization of several cell surface receptors and ion channels (*Almedom et al., 2009*; *Gottschalk et al., 2005*; *Kamat et al., 2014*; *Rosemond et al., 2011*). More recently, all four genes were identified in a genome-wide screen for factors that impair surface expression of a mutant TRP channel (*Talbot et al., 2019*). That these proteins can be stably isolated with TMCO1-bound ribosome-Sec61 complexes suggests a link between these observations and a co-translational process at the ER.

## Architecture of a ribosome-bound TMCO1 translocon

We sought to clarify the role of these ribosome-associated membrane components by examining their arrangement relative to key functional domains of the ribosome and the Sec61 complex. Using chemical cross-linking and mass spectrometry (XL-MS), we identified 1229 unique, high-confidence intra- and inter-protein cross-links in the affinity-purified complexes (*Figure 2—figure supplement 1A,B*). Multiple cross-links between 60S ribosomal subunits and the cytosolic-facing regions of Sec61, TMCO1, CCDC47, TMEM147 and NOMO confirmed their predicted membrane topologies, and placed them in the vicinity of the ribosome exit tunnel (*Figure 1D* and *Figure 2—figure supplement 1C,D*).

We next used single particle cryo-electron microscopy (cryo-EM) to directly visualize the natively purified complexes (*Figure 2*, *Figure 2—figure supplements 2–5* and *Table 1*). Density for the ribosome is well-defined, revealing hybrid state A/P and P/E tRNAs, and a mixture of nascent polypeptides in the exit tunnel (*Figure 2—figure supplement 5A*). Additional density is visible surrounding the ribosome exit tunnel. Local resolution within the translocon varies from ~3.5–4.5 Å in Sec61 and regions contacting the ribosome, to ~5.5–7.5 Å for most of the membrane region, and ~10–15 Å in peripheral and luminal regions (*Figure 2—figure supplement 4* and *Figure 2—figure supplement 5B,C*). Sec61 is in a conformation similar to that observed in the ribosome-Sec61-OST complex (*Braunger et al., 2018*), with a closed lateral gate and the plug helix occluding the central pore (*Figure 2—figure supplement 5D*).

A cluster of eight TMDs visible near the Sec61 hinge were unambiguously assigned to the Nicalin-TMEM147 sub-complex using a homology model based on the APH1-Nicastrin sub-complex of human γ-secretase (*Figure 2—figure supplement 6*; *Dettmer et al., 2010*; *Bai et al., 2015*; *Haffner et al., 2004*). The distinctive arrangement of the seven TMEM147 TMDs could be docked into the density as a rigid body with only minor adjustments (*Figure 2C* and *Figure 2—figure supplement 5E*). This enabled assignment of the remaining helical density to the single Nicalin TMD, which packs against TM1 of TMEM147 in its evolutionarily predicted position. Here, the large luminal domain of Nicalin extends into low-resolution density directly below the translocon (*Figure 2D*). Notably, the cytosolic end of TM3 in TMEM147 is ten residues shorter than the corresponding TM3 in APH-1 (*Figure 2—figure supplement 6E*). This allows TMEM147 to bind Sec61 despite limited space in the ribosome-translocon junction, and positions the short cytosolic regions of TMEM147 in contact with uL24 and rRNA H7, in agreement with the XL-MS (*Figure 2C* and *Figure 2—figure supplement 1C,D*).

Adjacent to the Nicalin-TMEM147 sub-complex is a cluster of three TMDs, which were assigned to TMCO1 (*Figure 2E*). We generated a model of TMCO1 using RaptorX-Contact (*Wang et al., 2017*; *Xu, 2019*), which employs co-evolutionary data and deep learning for distance-based structure prediction (*Figure 2—figure supplement 7*). The model recapitulates the conserved three TMD core, N-out/C-in topology, and cytosolic-facing coiled-coil found in members of the Oxa1 superfamily (*Anghel et al., 2017*; *Borowska et al., 2015*; *Kumazaki et al., 2014*), and could be placed into density with only minor adjustments. The conserved and positively charged coiled-coil of TMCO1 extends out of the membrane into the cytosolic vestibule where it packs against a surface on the ribosome that includes rRNA H19, H24 and uL24. This agrees with the proposed ribosome-binding mode of bacterial YidC (*Kedrov et al., 2016*), satisfies numerous inter- and intra-molecular cross-

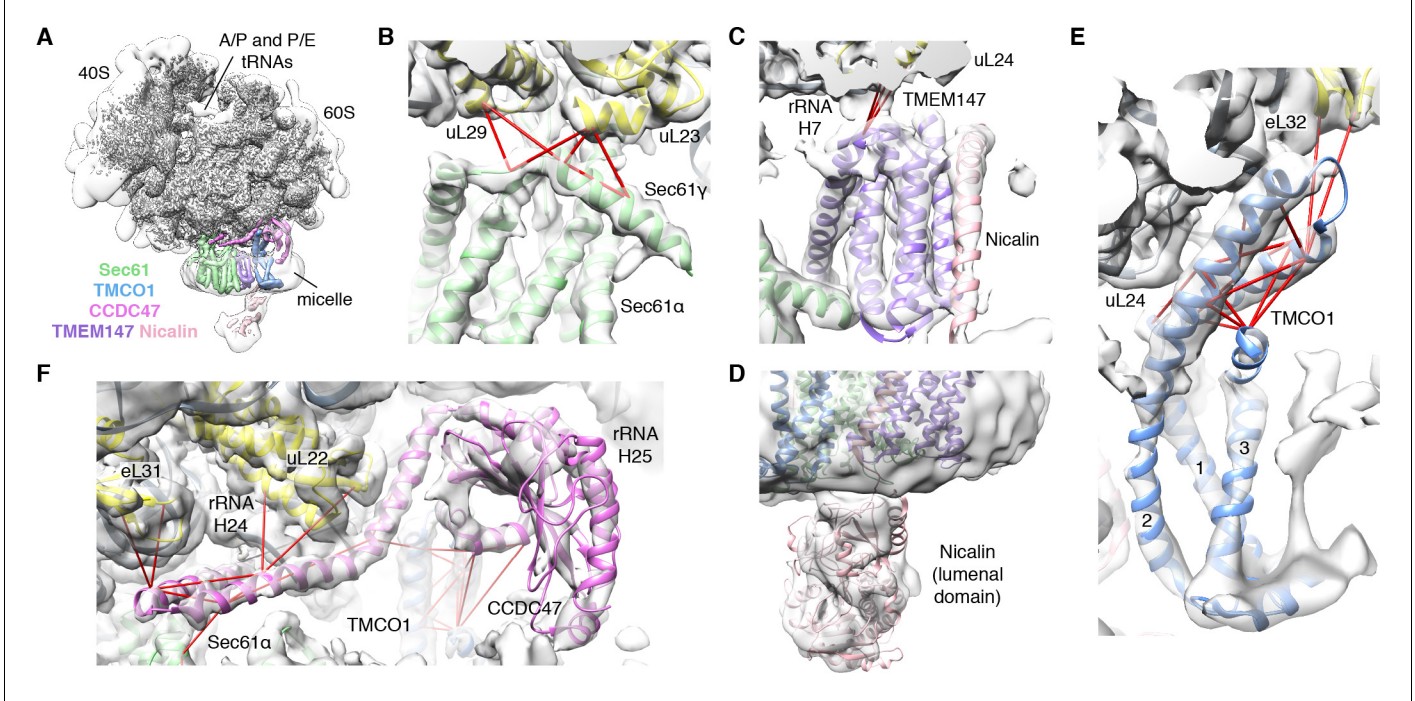

**Figure 2.** CryoEM structure of the ribosome-TMCO1 translocon complex. (**A**) Density for the 80S ribosome, A/P and P/E tRNAs is from the sharpened global map after low-pass filtering by local resolution. The translocon density is from the unsharpened focused map after low-pass filtering by local resolution; isolated densities for Sec61 (green), TMEM147 (purple), TMCO1 (blue) and CCDC47 (violet) are shown at a single threshold. The focused map is also shown at a lower threshold (transparent) to highlight luminal density and the micelle. (**B**) Closeup of the Sec61 complex, including experimentally observed cross-links (red) between Sec61γ and the indicated ribosomal subunits (yellow). (**C**) Closeup of the TMEM147-Nicalin complex (purple, pink), and cross-links between uL24 and the conserved TM3-TM4 loop of TMEM147. (**D**) The luminal domain of Nicalin (pink) extends below TMEM147 in a large lobe of density. (**E**) Closeup of TMCO1, and multiple intra- and inter-molecular cross-links. (**F**) Closeup of the cytosolic domain of CCDC47 and cross-links to the indicated ribosomal subunits, Sec61α, and the TMCO1 coiled coil; a cross-link that exceeds the distance cutoff of 35 Å is in black. Density in panels B-F is from the unsharpened signal-subtracted map after low-pass filtering by local resolution.

The online version of this article includes the following figure supplement(s) for figure 2:

**Figure supplement 1.** Cross-linking and mass spectrometry analysis of TMCO1-associated ribosomes.
**Figure supplement 2.** Representative cryo-EM image and 2D class averages for the natively purified ribosome-TMCO1 translocon complex.
**Figure supplement 3.** Cryo-EM data processing workflow.
**Figure supplement 4.** Resolution estimates for the TMCO1-ribosome complex.
**Figure supplement 5.** Additional views of local map and model quality.
**Figure supplement 6.** iTasser homology modeling of the human TMEM147-Nicalin complex.
**Figure supplement 7.** RaptorX-Contact modeling of human TMCO1.
**Figure supplement 8.** RaptorX-Contact modeling of human CCDC47.

links observed by XL-MS (*Figure 2E* and *Figure 2—figure supplement 1C,D*), and rationalizes the previously reported ribosome-binding activity of TMCO1 (*Anghel et al., 2017*).

The most prominent feature in the cytosolic vestibule is a globular density that curls out of the membrane near TMCO1 and terminates in a long helical extension that traces along the ribosome surface (*Figure 2F*). Using RaptorX-Contact we generated a model of the large cytosolic region of CCDC47, revealing a long and flexible C-terminal coiled-coil extending from a globular N-terminal domain (*Figure 2—figure supplement 8*). After placing the globular domain as a rigid body, the coiled-coil was adjusted to fit the extended helical density. The globular domain of CCDC47 contacts eL6 and rRNA H25, while the conserved and positively charged coiled-coil wedges between Sec61 and rRNA H24, before terminating at the mouth of the exit tunnel. This satisfies multiple intra- and inter-molecular cross-links to Sec61α, to the flexible N-terminus of Sec61β, to uL22, eL31 and eL32, and to the TMCO1 coiled-coil and C-terminal helix (*Figure 2F* and *Figure 2—figure supplement 1C,D*).

**Table 1.** Cryo-EM data collection and refinement statistics.

**Data collection and processing**

| Magnification | 64,000 | | |
|---|---|---|---|
| Voltage (kV) | 300 | | |
| Electron exposure (e⁻/Å²) | 50 | | |
| Defocus range (μm) | −1.0 to −2.5 | | |
| Pixel size (Å) | 0.68 | | |
| Symmetry imposed | C1 | | |
| Micrographs used | 5562 | | |
| Initial particle images (no.) | 1,049,128 | | |
| Final particle images (no.) | 82,684 | | |
| | Map 1 (EMD-21426) | Map 2 (EMD-21427) | Map 3 (EMD-21435) |
| Map resolution (Å) | 3.8 | 3.4 | 3.8 |
| FSC threshold | 0.143 | 0.143 | 0.143 |
| Map resolution range (Å) | 3.3 to 10.3 | 3.0 to 15.4 | 3.2 to 13.1 |

| | 60S–translocon PDB ID 6W6L |
|---|---|
| Refinement and validation | |
| Resolution for refinement (Å) | 3.8 |
| FSC threshold | 0.143 |
| Model composition | |
| Protein residues | 8012 (1718)* |
| Nucleotide bases | 3939 (0) |
| Average B factors (Å²) | |
| Protein | 156 (260) |
| Nucleotide | 191 (N/A) |
| R.M.S. deviations | |
| Bond lengths (Å) | 0.006 (0.005) |
| Bond angles (°) | 0.809 (1.013) |
| Validation | |
| MolProbity score | 2.03 (2.43) |
| Clash score | 9.41 (13.66) |
| Rotamer outliers | 1.42 (3.39) |
| Ramachandran plot | |
| Favored (%) | 93.7 (94.4) |
| Allowed (%) | 6.2 (5.4) |
| Outliers (%) | 0.1 (0.2) |

*Values in parentheses are for the translocon components only.

The translocon extends ~90 x 120 x 140 Å, with Sec61 and its accessory factors arranged near the ribosome exit tunnel and significant mass on both sides of the membrane (*Figure 3A*). Ribosome binding is mediated by multiple protein-protein and protein-RNA contacts (*Figure 3A,C*). These position the cytosolic domains of TMCO1 and CCDC47 near the nascent chain as it emerges from the ribosome exit tunnel, and place the luminal Nicalin domain near translocated segments of the nascent chain. Notably, the long C-terminal coiled-coil of CCDC47 extends to the nascent chain at the ribosome exit tunnel (*Figure 3A,B*). Truncating this conserved motif causes a developmental disorder in humans (*Morimoto et al., 2018*), suggesting that this interaction is functionally important.

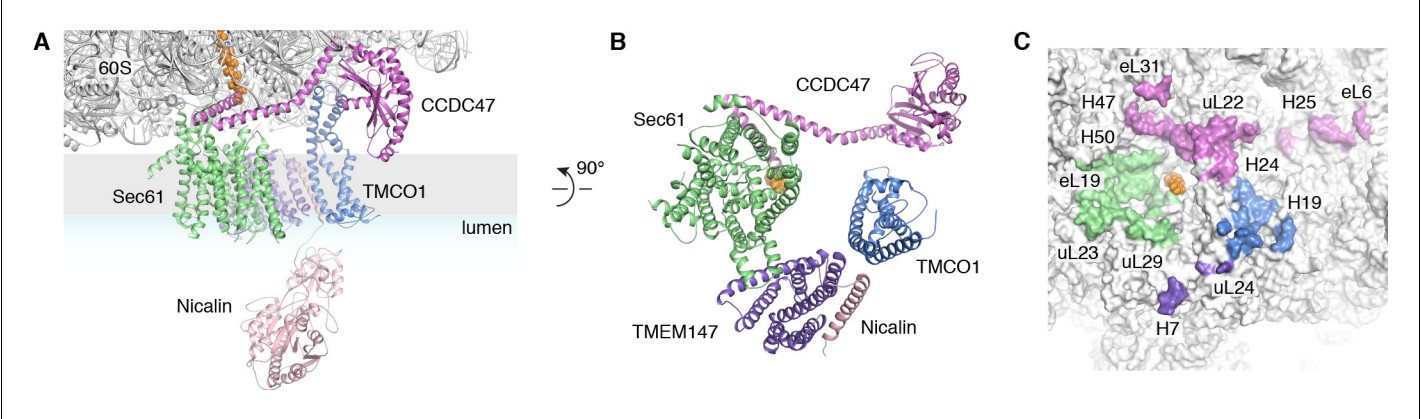

**Figure 3.** Organization of the TMCO1 translocon around the ribosome exit tunnel. (**A**) Closeup view showing TMCO1 (blue), CCDC47 (violet), TMEM147 (purple), Nicalin (pink) and the Sec61 complex arranged near the nascent polypeptide (orange spheres) at the mouth of the ribosome exit tunnel. (**B**) View of the translocon from the membrane (the Nicalin luminal domain was omitted for clarity). (**C**) Surface representation of the ribosome large subunit showing regions that contact Sec61, TMEM147, TMCO1 and CCDC47.

A prominent feature of the complex is a large (~25 x 30 x 30 Å), lipid-filled cavity formed at the center of the translocon by Sec61, TMEM147 and TMCO1 (*Figure 3B*). Like other structurally characterized members of the Oxa1 superfamily (*Anghel et al., 2017*; *Kumazaki et al., 2014*; *Borowska et al., 2015*), the transmembrane helices of TMCO1 form a funnel that extends from the cytosol into the lipid bilayer (*Figure 4A*). In bacterial YidC, this funnel operates as a transient binding site for TMDs, which are then released into the membrane (*Kumazaki et al., 2014*; *Borowska et al., 2015*). TMCO1 is located on the 'back' side of Sec61 in the TMCO1 translocon (*Figure 4B*). Here, the TMCO1 funnel lines the lipid-filled cavity at the center of the translocon, suggesting that a hydrophobic segment could be inserted from the cytosol into a protected membrane environment.

TMEM147 also lines the lipid-filled cavity. Here, its seven TMDs form a funnel that extends from the lumen partway across the membrane. Within the bilayer, the Sec61 hinge (located between TM5 and TM6) contacts TM2, TM3 and TM4 inside the TMEM147 funnel. A similar intra-membrane interaction is observed in γ-secretase, where the presenilin C-terminus fills the hydrophobic APH-1 funnel (*Figure 4—figure supplement 1B*; *Bai et al., 2015*). Unlike in γ-secretase, however, the Sec61 hinge only partially occupies the TMEM147 funnel, laterally sealing it in membrane, but leaving it open to the lumen (*Figure 4C,D*). This is reminiscent of the Hrd1 protein conducting ERAD channel, in which a structurally similar hydrophilic funnel, proposed to transport transmembrane segments from the bilayer to the cytosol, opens to the cytosol, and is laterally sealed by a neighboring Hrd1 subunit (*Figure 4—figure supplement 1C*; *Schoebel et al., 2017*). By analogy, TMEM147 could insert a hydrophobic segment from the lumen into the central membrane cavity in a process gated by Sec61. Taken together, these structural observations suggest that the TMCO1 translocon may be specialized for membrane protein biogenesis.

## The TMCO1 translocon functions in multi-pass membrane protein biogenesis

We sought to test this possibility by sequencing the mRNAs associated with ribosomes recovered after affinity purification via the Flag tag on TMCO1 (RIP-seq). Remarkably, we observed strong enrichment for transcripts encoding secretory pathway transmembrane proteins (*Figure 5A*). Of these, single-pass proteins—by far the most abundant type of membrane protein in the human genome—were strongly depleted (*Figure 5B*). By contrast, transcripts encoding multi-pass membrane proteins with four or more TMDs were enriched (*Figure 5B*). These include numerous transporters, receptors, transferases and hydrolases (*Figure 5C*). Consistent with selective enrichment of TMCO1-linked transcripts, we observed enrichment across three orders of magnitude of transcript abundance in the input sample (*Figure 5D*), and this was independent of protein length (*Figure 5—*

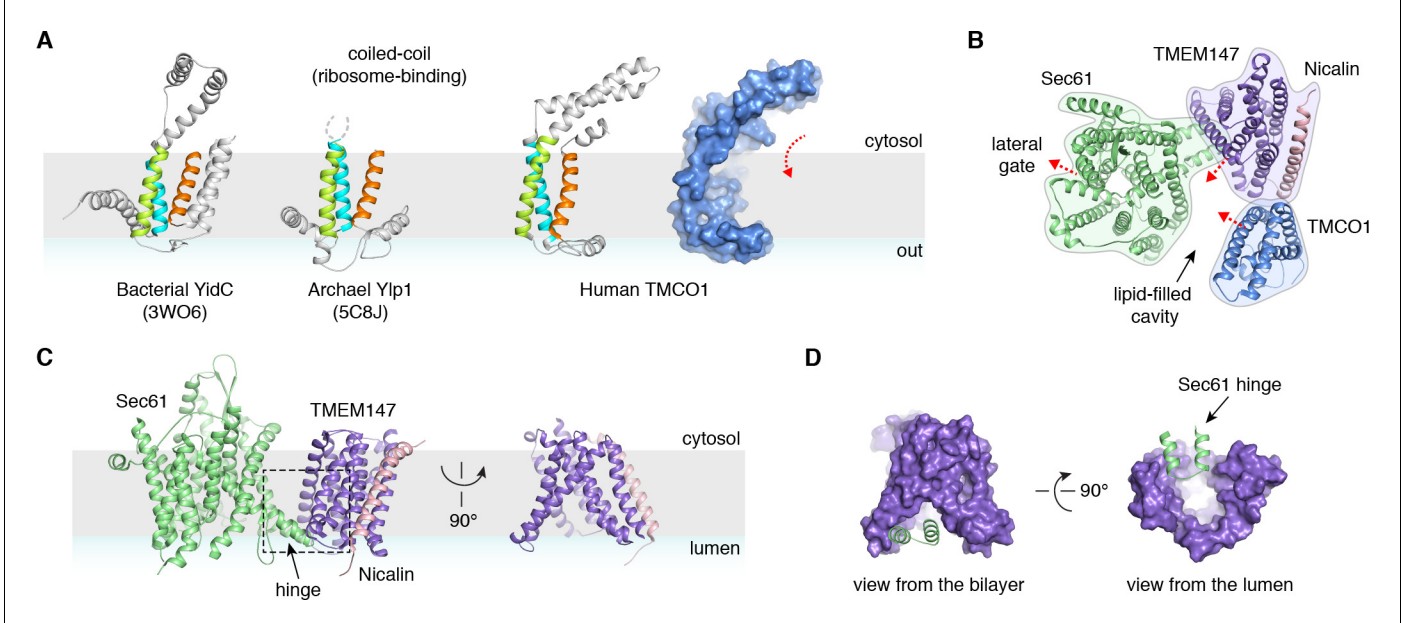

**Figure 4.** Conserved structural features suggest pathways into the membrane. (**A**) Comparison of experimentally determined structures for bacterial, archaeal and human members of the Oxa1 superfamily of membrane protein biogenesis factors. The evolutionarily conserved three TMD core (cyan, lime and orange), forms a funnel extending from the cytosol into the bilayer. A surface representation of TMCO1 (blue) is shown at right. (**B**) Slice through the membrane of the TMCO1 translocon, viewed from the cytosol. The large lipid-filled cavity formed by Sec61, TMEM147 and TMCO1 is visible at the center of the translocon. Red arrows indicate known (via Sec61) and potential (via TMCO1 and TMEM147) routes into the membrane. (**C**) Closeup of the Sec61-TMEM147 interaction. The seven TMDs of TMEM147 (purple) form a large funnel that extends from the ER lumen into the lipid bilayer. (**D**) Surface representation of TMEM147. The hinge region of Sec61 (green) partially occludes the bilayer-exposed opening of the funnel (left), but not the luminal opening (right).

The online version of this article includes the following figure supplement(s) for figure 4:

**Figure supplement 1.** Structural comparisons with γ-secretase and the Hrd1 protein conducting ERAD channel.
**Figure supplement 2.** Structural comparison of the TMCO1- and OST translocons.

---

*figure supplement 1A*). These data directly link the TMCO1 translocon to a co-translational process involving hundreds of different multi-pass clients.

To evaluate the role of the TMCO1 translocon in biogenesis, we monitored the endogenous protein levels of the 'Excitatory amino acid transporter 1' (EAAT1; SLC1A3; GLAST-1) in HEK293 cells lacking different accessory components. EAAT1 is a member of the large solute carrier (SLC) transporter superfamily, more than one-third of which were enriched by RIP-seq. EAAT1 functions as a homotrimer, and its structure contains multiple TMDs of marginal hydrophobicity and re-entrant helical loops on both sides of the membrane, all of which are required for function (*Canul-Tec et al., 2017*).

Compared to wild-type cells, the steady-state expression level of EAAT1 was reduced by ~3 fold in *TMCO1* knockout cells, but was unaffected in cells lacking the auxiliary translocon component TRAM (*Figure 5E–G*). Similar reductions were observed in *Nicalin* (2.4-fold), *TMCO1/Nicalin* (2.8-fold) and *TMCO1/CCDC47* (3.8-fold) single- and double-knockout cells, while a single *CCDC47* (1.6-fold) knockout showed only a modest reduction (*Figure 5F,G*). By contrast, the steady-state expression levels and glycosylation patterns of two single-pass membrane proteins, integrin α5 and TRAPα, were unchanged, demonstrating that TMCO1 disruption does not lead to a general defect in membrane protein biogenesis (*Figure 5—figure supplement 1B*). We also observed little change in EAAT1 mRNA levels in *TMCO1*, *Nicalin* and *CCDC47* single- and double-knockout cells. Together with the structural analysis, these data implicate the TMCO1 translocon in multi-pass membrane protein biogenesis.

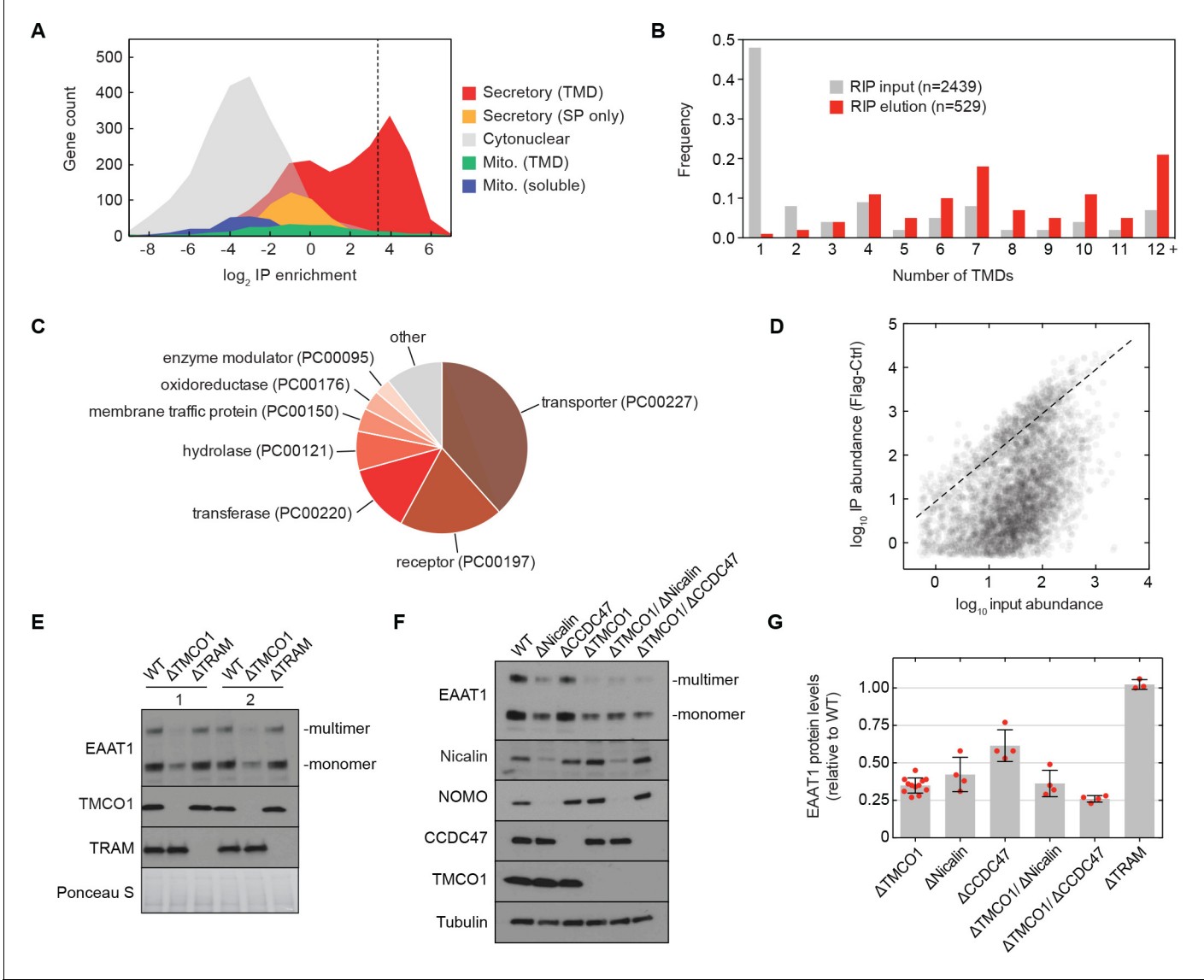

**Figure 5.** The TMCO1 translocon acts on multi-pass membrane proteins. (**A**) Log2 enrichment of transcripts encoding proteins of the indicated categories according to Uniprot annotation. Enrichment was calculated as (Flag IP - Ctrl IP)/Input, where "Flag IP" and "Ctrl IP" are the average transcript levels in the ribosome fraction following anti-Flag immunoprecipitation from digitonin-solubilized wild-type (Ctrl) or 3xFlag-TMCO1 (Flag) HEK293 membranes (n=3), and "Input" is the average transcript abundance in the total membrane fraction (n=2). More than 98% of the most enriched transcripts (right of the dashed line) encode secretory pathway transmembrane proteins. (**B**) Proportion of secretory pathway transmembrane proteins containing the indicated number of Uniprot-predicted TMDs in the input (gray), and in the 529 most enriched membrane-protein encoding transcripts from the elution (red). (**C**) PANTHER classification for the enriched set of membrane proteins. (**D**) Transcript levels in the TMCO1 immunoprecipitated sample ("IP abundance") plotted against transcript levels in total HEK293 membranes ("input abundance"). Enrichment (above the dashed line) is seen across three orders of magnitude of input mRNA abundance. (**E**) Representative western blot of total HEK293 lysate from wild-type (WT) and knockout (ΔTMCO1 and ΔTRAM) cells, in duplicate. Expression of the multi-pass membrane protein EAAT1, is decreased in the TMCO1 knockout cells, but is unaffected by deletion of the unrelated Sec61 accessory factor, TRAM. Note that the EAAT1 "multimer" band is from SDS-induced aggregation. (**F**) As in (**E**), for the indicated single- and double-knockout cells. Tubulin serves as a loading control. Note that disruption of Nicalin reduces the expression levels of its binding partner, NOMO, as shown previously (*Dettmer et al., 2010*). The faint band migrating just above Nicalin in the ΔNicalin cells is a cross-reacting band. (**G**) Quantification of protein expression levels for EAAT1 (monomer and multimer), showing mean and S.D., relative to WT cells. The online version of this article includes the following figure supplement(s) for figure 5:

**Figure supplement 1.** Additional functional analysis.

## Discussion

Our data identify TMCO1, CCDC47 and the Nicalin-TMEM147-NOMO complex as conserved components of an ER translocon that functions co-translationally with Sec61 during biogenesis of multi-pass membrane proteins. The biochemical function of the TMCO1 translocon is currently unclear. Although we do not formally exclude a role in client-specific targeting to the ER, we propose that the TMCO1 translocon functions as an insertase and intramembrane chaperone. Our structural model for TMCO1 is consistent with its evolutionary relationship to members of the Oxa1 superfamily, including YidC, Get1, EMC3 and Ylp1 (*Anghel et al., 2017*). These proteins have evolved to function in different contexts, but their ability to move transmembrane segments into the membrane appears to be conserved (*Wang et al., 2014*; *Klenner et al., 2008*; *Pleiner et al., 2020*; *Yu et al., 2008*). By analogy, we propose that hydrophobic segments of the nascent chain that inefficiently engage with Sec61 could access the membrane through the conserved cytosolic TMCO1 funnel. In addition, hydrophobic segments that have translocated across the bilayer through the canonical Sec61 channel might access the membrane through the luminal TMEM147 funnel. As segments integrate, the central cavity of the translocon could shield the nascent chain to minimize misfolding and degradation. Organizing these putative functions in a single translocon might increase the efficiency with which different biophysical and topological features of the nascent chain are accommodated during multi-pass membrane protein biogenesis (*Cymer et al., 2015*; *Lu et al., 2000*; *Skach, 2009*). High resolution structures of the TMCO1 translocon and analysis of its interactions with substrate will be important for testing this model.

More broadly, our data support a general view of the translocon as a dynamic assembly whose subunit composition varies temporally to meet the demands of a particular client (*Conti et al., 2015*; *Johnson and van Waes, 1999*). TMCO1, CCDC47 and the Nicalin-TMEM147-NOMO complex are abundant (*Itzhak et al., 2016*), which presumably allows them to compete with other translocon-associated factors for access to the nascent chain. Indeed, while many multi-pass clients of the TMCO1 translocon harbor N-terminal STT3A glycosylation sites (*Cherepanova et al., 2019*), we see little biochemical or structural evidence for an associated OST complex. This is consistent with the substantial steric overlap observed between the membrane and luminal regions of OST and the TMCO1 accessory factors (*Figure 4—figure supplement 2*), which likely dictates that they alternately access the nascent chain during synthesis. Notably, TMCO1, CCDC47 and the Nicalin-TMEM147-NOMO complex do not stably associate in the absence of ribosomes (*Figure 1—figure supplement 1*). An intriguing possibility is that these are modular components, capable of acting independently in additional contexts.

A general role for the TMCO1 translocon in multi-pass membrane protein biogenesis is consistent with the wide expression and conservation of its subunits, and the numerous cellular and organismal phenotypes associated with their dysfunction. In humans, TMCO1 has been linked to glaucoma (*Burdon et al., 2011*; *Sharma et al., 2012*), and loss of either TMCO1 (*Xin et al., 2010*; *Caglayan et al., 2013*; *Alanay et al., 2014*) or CCDC47 (*Morimoto et al., 2018*) causes rare autosomal recessive developmental disorders. Similarly, a mutation in TMEM147 has been linked to a rare neurodevelopmental disorder manifesting with severe intellectual disability and impaired vision (*Reuter et al., 2017*).

At the cellular level, disrupting TMCO1, CCDC47, Nicalin, TMEM147 or NOMO leads to reduced fitness (*Wang et al., 2015*). Cells lacking CCDC47 show attenuated ERAD (*Yamamoto et al., 2014*) and impaired $Ca^{2+}$ signaling (*Zhang et al., 2007*; *Konno et al., 2012*), while the Nicalin-TMEM147-NOMO complex is linked to Nodal signaling (*Haffner et al., 2004*) and altered localization and subunit composition of some multi-pass membrane proteins (*Almedom et al., 2009*; *Kamat et al., 2014*; *Rosemond et al., 2011*). Cells lacking TMCO1 show defects in $Ca^{2+}$ handling, which has led to the proposal that TMCO1 functions as a $Ca^{2+}$-channel (*Wang et al., 2016*). Our data reconcile these different observations, which likely result from biogenesis defects in hundreds of different multi-pass proteins.

As the folding capacity of the cell must be robust to mutations and other stresses that affect folding efficiency, it is likely that other ER chaperones and accessory factors can partially compensate for loss of TMCO1 translocon components. In this regard, it will be important to define the functional relationship between the TMCO1 translocon and the ER membrane complex (EMC), each of which harbors a subunit belonging to the Oxa1 superfamily (*Anghel et al., 2017*)

and facilitates multi-pass membrane protein biogenesis (*Chitwood et al., 2018*; *Guna et al., 2018*; *Shurtleff et al., 2018*; *Volkmar et al., 2019*; *Tian et al., 2019*).

# Materials and methods

## Key resources table

| Reagent type (species) or resource | Designation | Source or reference | Identifiers | Additional information |
|---|---|---|---|---|
| Antibody | Anti-FLAG M2 Affinity Gel (mouse monoclonal) | Sigma | Cat# A2220, RRID:AB_10063035 | |
| Antibody | Rabbit polyclonal, FLAG | Sigma | Cat# F7425, RRID:AB_439687 | WB (1:1000) |
| Antibody | Rabbit, polyclonal TMCO1 | *Anghel et al., 2017* | | WB (1:1000) |
| Antibody | Rabbit, polyclonal Sec61β | *Görlich et al., 1992* | | WB (1:10000) |
| Antibody | Mouse, monoclonal EAAT1 | Santa Cruz | Cat# sc-515839 | WB (1:1000) |
| Antibody | Rabbit, polyclonal Sec61α | Thermo Fisher | Cat# PA5-21773, RRID:AB_11152794 | WB (1:1000) |
| Antibody | Rabbit, polyclonal L17 | Abgent | Cat# AP9892b, RRID:AB_10613776 | WB (1:1000) |
| Antibody | Mouse monoclonal STT3A | Novus | Cat# H00003703-M02, RRID:AB_2198043 | WB (1:1000) |
| Antibody | Mouse monoclonal Tubulin | Abcam | Cat# ab7291, RRID:AB_2241126 | WB (1:1000) |
| Antibody | Rabbit polyclonal Integrin α5 | Cell Signaling | Cat# 4705, RRID:AB_2233962 | WB (1:1000) |
| Antibody | Rabbit polyclonal Nicalin | Bethyl | Cat# A305-623A-M, RRID:AB_2782781 | WB (1:1000) |
| Antibody | Rabbit polyclonal TMEM147 | Thermo Fisher | Cat# PA5-95876, RRID:AB_2807678 | WB (1:1000) |
| Antibody | Goat polyclonal Nomo1 | Thermo Fisher | Cat# PA5-47534, RRID:AB_2607776 | WB (1:1000) |
| Antibody | Rabbit polyclonal CCDC47 | Bethyl | Cat# A305-100A, RRID:AB_2631495 | WB (1:1000) |
| Cell line (*H. sapiens*) | Flp-In T-REx 293 | Thermo Fisher | Cat# R78007 | |
| Cell line (*H. sapiens*) | Flp-In T-REx 293, 3xFlag-Cas9 | *Anghel et al., 2017* | | 3xFlag-Cas9 integrated into FRT site |
| Cell line (*H. sapiens*) | Flp-In T-REx 293, 3xFlag-Cas9, 3xFlag-TMCO1 | *Anghel et al., 2017* | | Obtained by CRISPR-Cas9; one nonfunctional and one N-terminally tagged TMCO1 allele |

*Continued on next page*

*Continued*

| Reagent type (species) or resource | Designation | Source or reference | Identifiers | Additional information |
|---|---|---|---|---|
| Cell line (*H. sapiens*) | Flp-In T-REx 293, 3xFlag-Cas9, ΔTMCO1 | *Anghel et al., 2017* | | TMCO1 disrupted by CRISPR-Cas9 |
| Cell line (*H. sapiens*) | Flp-In T-REx 293, 3xFlag-Cas9, ΔNicalin | This paper | | Nicalin disrupted by CRISPR-Cas9 |
| Cell line (*H. sapiens*) | Flp-In T-REx 293, 3xFlag-Cas9, ΔCCDC47 | This paper | | CCDC47 disrupted by CRISPR-Cas9 |
| Cell line (*H. sapiens*) | Flp-In T-REx 293, 3xFlag-Cas9, ΔTMCO1, ΔNicalin | This paper | | Nicalin disrupted by CRISPR-Cas9 in ΔTMCO1 background |
| Cell line (*H. sapiens*) | Flp-In T-REx 293, 3xFlag-Cas9, ΔTMCO1, ΔCCDC47 | This paper | | CCDC47 disrupted using CRISPR-Cas9 in ΔTMCO1 background |
| Cell line (*H. sapiens*) | Flp-In T-REx 293, ΔTMCO1 | This paper | | TMCO1 disrupted by CRISPR-Cas9 |
| Cell line (*H. sapiens*) | Flp-In T-REx 293, ΔTRAM | This paper | | TRAM1 disrupted by CRISPR-Cas9 |
| Cell line (*H. sapiens*) | Flp-In T-REx 293, 3xFlag-Cas9, ΔNicalin, 3xFlag-Nicalin | This paper | | Randomly integrated 3xFlag-Nicalin in ΔNicalin background |
| Cell line (*H. sapiens*) | Flp-In T-REx 293, 3xFlag-Cas9, ΔTMCO1, 3xFlag-TMCO1 | This paper | | Randomly integrated 3xFlag-TMCO1 in ΔTMCO1 background |
| Recombinant DNA reagent | pEGFP-n1 | Addgene | Cat# 6085–1 | |
| Recombinant DNA reagent | pEGFP-3xFlag-TMCO1 | This paper | | Human TMCO1 with an N-terminal 3xFlag tag |
| Recombinant DNA reagent | pEGFP-3xFlag-Nicalin | This paper | | Human Nicalin with an N-terminal 3xFlag tag following the signal peptide |
| Recombinant DNA reagent | pX330 | Addgene | Cat# 42230 | |
| Recombinant DNA reagent | pX330-TRAM1-sgRNA | This paper | | TTTGATGCCATAGTAATAAA |
| Sequence-based reagent | sgRNA targeting Nicalin | Invitrogen | Custom Synthesis | ACGGAATGCAGTGCTGAACA |
| Sequence-based reagent | sgRNA targeting CCDC47 | Invitrogen | Custom Synthesis | TCAGTGATTATGACCCGTT |
| Sequence-based reagent | GAPDH fwd | IDT | Custom Synthesis | ACAACTTTGGTATCGTGGAAGG |
| Sequence-based reagent | GAPDH rev | IDT | Custom Synthesis | GCCATCACGCCACAGTTTC |

*Continued on next page*

Continued

| Reagent type (species) or resource | Designation | Source or reference | Identifiers | Additional information |
|---|---|---|---|---|
| Sequence-based reagent | EAAT1 fwd | IDT | Custom Synthesis | TTCCTGGGGAACTTCTGATG |
| Sequence-based reagent | EAAT1 rev | IDT | Custom Synthesis | CCATCTTCCCTGATGCCTTA |
| Peptide, recombinant protein | 3xFlag Peptide | ApexBio | Cat# A6001 | |
| Peptide, recombinant protein | micrococcal nuclease | NEB | Cat# M0247S | |
| Peptide, recombinant protein | DNAseI | Promega | Cat# M6101 | |
| Peptide, recombinant protein | EndoH | NEB | Cat# P0702 | |
| Peptide, recombinant protein | PNGaseF | Promega | Cat# 9PIV483 | |
| Commercial assay or kit | iScript gDNA Clear cDNA Synthesis Kit | Bio-Rad | Cat# 1725034 | |
| Commercial assay or kit | iTaq Universal SYBR Green Supermix | Bio-Rad | Cat# 1725120 | |
| Commercial assay or kit | RiboZero | Illumina | Cat# 20037135 | |
| Commercial assay or kit | Universal Mycoplasma Detection Kit | ATCC | Cat# 30–1012K | |
| Chemical compound, drug | digitonin | Calbiochem | Cat# 11024-24-1 | |
| Chemical compound, drug | disuccinimidyl suberate | Thermo Fisher | Cat# 21555 | |
| Chemical compound, drug | TRIzol | Ambion | Cat# 15596018 | |
| Software, algorithm | RELION, v.3.1 | *Zivanov et al., 2018* | RRID:SCR_016274 | |
| Software, algorithm | MotionCor2 | *Zheng et al., 2017* | RRID:SCR_016499 | |
| Software, algorithm | GCTF v.0.5 | *Zhang, 2016* | RRID:SCR_016500 | |
| Software, algorithm | Protein Prospector v.5.23.0 | *Trnka et al., 2014* | RRID:SCR_014558 | |
| Software, algorithm | Proteome Discoverer v.2.2 | Thermo Scientific | RRID:SCR_014477 | |
| Software, algorithm | UCSF Chimera v.1.13.1 | *Pettersen et al., 2004* | RRID:SCR_004097 | |

*Continued on next page*

*Continued*

| Reagent type (species) or resource | Designation | Source or reference | Identifiers | Additional information |
|---|---|---|---|---|
| Software, algorithm | Pymol v.2.3 | www.pymol.org | RRID:SCR_000305 | |
| Software, algorithm | Phenix v.1.18–3845 | *Afonine et al., 2018* | RRID:SCR_014224 | |
| Software, algorithm | RaptorX-Contact | *Wang et al., 2017* | RRID:SCR_018118 | |
| Software, algorithm | i-Tasser | *Zhang, 2008* | RRID:SCR_014627 | |
| Software, algorithm | SBGrid | *Morin et al., 2013* | RRID:SCR_003511 | |
| Software, algorithm | Coot v.0.9 | *Emsley et al., 2010* | RRID:SCR_014222 | |
| Other | Quantifoil 1.2/1.3 200 mesh, pre-coated with amorphous 2 nm Carbon | Ted Pella, Inc | Cat# 668–200-CU | |
| Other | Freestyle 293 Expression media | Fisher Scientific | Cat# 12-338-026 | |
| Other | 1 L PETG square media bottles | Fisher Scientific | Cat# 09-923-16C | |

## Antibodies

Antibodies against human TMCO1, Sec61β and TRAPα were characterized previously (*Fons et al., 2003*; *Anghel et al., 2017*; *Görlich et al., 1992*). Additional antibodies were obtained from the following sources: anti-EAAT1 (Santa Cruz, sc-515839), anti-Sec61α (Thermo Fisher, PA5-21773), anti-uL22 (Abgent, AP9892b), anti-STT3A (Novus, H00003703-M02), anti-Tubulin (Abcam, ab7291), anti-Integrin α5 (Cell signaling, 4705) anti-Nicalin (Bethyl, A305-623A-M), anti-TMEM147 (Thermo Fisher, PA5-95876), anti-NOMO (Thermo Fisher, PA5-47534), anti-CCDC47 (Bethyl, A305-100A), anti-TRAM1 (Abcam, ab190982), anti-Mouse rabbit HRP (Abcam, ab6708), anti-Rabbit donkey HRP (Sigma, SAB3700863), anti-Goat rabbit HRP (Sigma, A5420).

## Cell culture

Flp-In T-REx 293 cells containing a 3xFlag-Cas9 construct were maintained in DMEM supplemented with 10% FBS (Gemini Foundation) and penicillin/streptomycin mixture (Invitrogen). TMCO1 knockout and 3xFlag-TMCO1 HEK293 cell lines have been described and characterized previously (*Anghel et al., 2017*). Nicalin and CCDC47 knockout cell lines were generated using the CRISPR/Cas9 system, in both parental and TMCO1 knockout backgrounds. Cas9 expression was induced by addition of 10 ng/mL doxycycline followed by transfection of sgRNAs targeting either Nicalin (ACGGAATGCAGTGCTGAACA) or CCDC47 (TCAGTGATTATGACCCGTT). Cells were grown for 48 hr, followed by single cell sorting into 96 well plates for clonal isolation. Nicalin and CCDC47 knockouts in parental and TMCO1 knockout backgrounds were verified by western blot and genomic DNA sequencing.

A TRAM1 knockout cell line was generated using the CRISPR/Cas9 system in Flp-In T-Rex 293 cells (Thermo Fisher) by transfecting a modified pX330 plasmid (Addgene) expressing human codon-optimized Cas9 and an sgRNA targeting TRAM1 (TTTGATGCCATAGTAATAAA). Single cells were isolated by sorting and allowed to grow clonally. The final TRAM1 knockout was verified by western blot and genomic DNA sequencing.

To scale up the sample preparation for crosslinking mass spectrometry (XL-MS) and cryo-EM, Flp-In TRex 293 cells expressing 3xFlag-TMCO1 from the endogenous TMCO1 promoter were cultured in suspension. Cells were grown in 1 L PETG square media bottles (Fisher, 09-923-16C) containing 250 ml Freestyle 293 media (Gibco) supplemented with 10 mM Hepes pH 7.5 (Invitrogen), 10 mM

L-glutamine (Invitrogen), 0.3 µg/ml penicillin (Gemini), 0.5 µg/ml streptomycin (Gemini), and 0.5% FBS (Gemini Foundation), at 37°C, 5% CO2, and 135 rpm, to a final density of ~$1\times10^6$ cells/ml.

Stable cell lines overexpressing N-terminally 3xFlag tagged TMCO1 and Nicalin were generated by transfecting the respective knockout cell lines with a modified pEGFP-n1 plasmid (Addgene) encoding N-terminally 3xFlag-tagged TMCO1 or Nicalin (tag inserted after the signal peptide), under the control of a CMV promoter. Cells were transfected using the TransIT-293 transfection reagent (Mirus) and selected for 14 days by treatment with 0.7 mg/ml G418 (Invitrogen), with selection media changed every 3 days. Selected cells were maintained in DMEM supplemented with 10% FBS, penicillin/streptomycin, and 0.3 mg/ml G418. Expression was verified by western blot.

Cells were checked approximately every three months for mycoplasma contamination using the Universal Mycoplasma Detection Kit (ATCC), and were found to be negative.

## Isolation of TMCO1-ribosome complexes for interaction analysis

For mass spectrometry, approximately $2 \times 10^8$ of wild-type (control) and 3xFlag-TMCO1 cells were pelleted, resuspended in ice cold hypotonic lysis buffer (10 mM Hepes pH 7.4, 10 mM potassium acetate, 1 mM magnesium chloride) and incubated on ice for 15 min. Unless otherwise noted, all buffers included emetine at a final concentration of 50 µg/ml. Cells were lysed with 25 strokes of a pre-chilled dounce tissue grinder with a tight-fitting pestle, then 250 mM sucrose and 1 mM PMSF was added to the lysate. Nuclei were pelleted by centrifugation at 700 x g for 3 min. The membrane-containing supernatant was removed and put on ice. The pellet was washed with 1 ml ice cold assay buffer (50 mM Hepes pH 7.4, 250 mM sucrose, 250 mM potassium acetate, 10 mM magnesium chloride) and centrifuged again, and the supernatant combined with the membrane fraction. Membranes were sedimented at 10,000 x g for 10 min at 4°C and resuspended in assay buffer to an $A_{260}$ of ~50.

Monosomes were generated by treating the resuspended membranes with 1 mM calcium acetate and 10,000 U of micrococcal nuclease (NEB, M0247S), and incubating at 25°C for 10 min. Nuclease activity was stopped by adding EGTA to a final concentration of 2 mM. Membranes were solubilized in ice cold assay buffer supplemented with 2.5% digitonin (Calbiochem 11024-24-1) for 15 min on ice, and insoluble material was removed by centrifugation at 10,000 x g for 10 min at 4°C.

TMCO1-ribosome complexes were affinity purified by incubating solubilized material with M2 Flag affinity gel (Sigma, A2220) for 1 hr at 4°C with gentle end-over-end mixing. Unbound material was removed by centrifugation, the resin was washed twice with five bed volumes of ice-cold wash buffer (50 mM Hepes pH 7.4, 250 mM sucrose, 350 mM potassium acetate, 10 mM magnesium chloride, 0.25% digitonin), and twice with five bed volumes of assay buffer supplemented with 0.25% digitonin. Bound material was eluted in two successive 30 min incubations with two bed volumes of ice-cold assay buffer supplemented with 0.25% digitonin and 0.5 mg/ml 3xFlag peptide, at 4°C with gentle end-over-end mixing. The ribosome containing fraction was obtained by sedimenting the IP elutions through a 1 mL sucrose cushion (1 M sucrose, 150 mM potassium chloride, 50 mM Tris pH 7.5, 5 mM magnesium chloride, 0.1% digitonin) at 250,000 x g for 2 hr in a TLA100.3 rotor.

Ribosome pellets were resuspended in 50 mM Hepes pH 7.4, 100 mM sodium chloride, 1% SDS. Proteins were then methanol-chloroform extracted, FASP trypsin-digested, TMT-labeled and analyzed in a single 180 min LC-MS/MS run at the Proteomics and Mass Spectrometry Facility at Harvard University. Enrichment ratios were calculated as Flag IP/control IP for all peptides identified more than once.

Small-scale IPs were performed similarly, using microsomes isolated from stably integrated 3xFlag-TMCO1 or 3xFlag-Nicalin HEK293 cells.

## Isolation of complexes for XL-MS and cryo-EM

Affinity purification of TMCO1-ribosome complexes for XL-MS and cryo-EM was done as described above with the following changes. Typically, XL-MS samples were produced from ~$4\times10^9$ cells, and cryo-EM samples from ~$7\times10^8$ cells. Emetine was not used. To remove any contaminating DNA, isolated membranes were treated with 5 U/ml RNase Free DNase (Promega, M6101) for 15 min at room temperature. Following affinity purification, TMCO1-ribosome complexes were isolated via sedimentation through a 300 µl sucrose cushion (0.5 M sucrose, 150 mM potassium acetate, 50 mM Hepes pH 7.4, 5 mM magnesium chloride, 0.25% digitonin) at 355,000 x g for 45 min in a TLA120.1

rotor. Pellets were resuspended in 150 mM potassium acetate, 50 mM Hepes pH 7.4, 5 mM magnesium chloride, 0.25% digitonin, and concentration determined by $A_{260}$.

## Chemical cross-linking and mass spectrometry

Approximately 85 µg of purified TMCO1-ribosome complexes were resuspended in 150 mM potassium acetate, 50 mM Hepes pH 7.4, 5 mM magnesium chloride, 0.25% digitonin to a concentration of 0.5 mg/ml. Crosslinking was performed by adding disuccinimidyl suberate (DSS, Thermo Fisher, 21555) (prepared as a fresh 10 mM stock in DMSO) to a final concentration of 0.5 mM and incubating for 30 min at 35°C. Crosslinking reactions were mixed by lightly agitating the tube every 5 min during incubation, and quenched by adding 100 mM Tris pH 8. Reactions were TCA precipitated before processing for mass spectrometry. Pellets were washed with ice cold acetone to remove excess lipid and detergent, and then pelleted again.

For mass spectrometry, the TCA precipitated material was resuspended in 8 M Urea and 10 mM TCEP, heated at 56°C for 20 min, alkylated with 15 mM iodoacetamide (30 min at room temperature), and then quenched with 15 mM dithiothreiotol (15 min at room temperature). The sample was then diluted to 2 M Urea and digested overnight with 1 µg trypsin (Promega Gold) for 4 hr at 37°C. A second aliquot of 1 µg trypsin was then added and digestion was allowed to proceed overnight. The digestion mixture was acidified to 0.5% TFA and diluted 6-fold prior to desalting on a Peptide C18 MacroTrap column (Michrom Bioresources) controlled by Akta Purifier (GE Healthcare Life Sciences) and evaporated to dryness. Crosslinked products were brought up in 10 µl of SEC buffer (70:30 $H_2O$:ACN with 0.1% TFA) and enriched by size-exclusion chromatography (Superdex Peptide, GE Healthcare Life Sciences) as in *Leitner et al., 2012*. 100 µl fractions eluting between 0.9 and 1.4 ml were dried, resuspended in 0.1% formic acid for MS analysis. The fractions starting at 0.9 ml and 1.3 ml were combined prior to evaporation to make four MS fractions.

Samples were reconstituted in 5 µl of 0.1% formic acid for mass spectrometry. LC-MS analysis was performed with an Orbitrap Fusion Lumos mass spectrometer (Thermo Scientific) coupled with a nanoelectrospray ion source (Easy-Spray, Thermo) and M-Class NanoAcquity UPLC system (Waters). Crosslink enriched fractions were separated on a 50 cm x 75 µm ID PepMap C18 column (Thermo). 2.5 µl of sample was loaded onto the column and eluted running a gradient from 3.5% solvent B (A: 0.1% formic acid in water, B: 0.1% in ACN) to 25% B in 175 min followed by a second gradient to 30% B over 10 min. Precursor scans were acquired in the Orbitrap from 375 to 1500 m/z (resolution: 120000, AGC Target: 4.0e5, max injection time: 50 ms). Precursor ions were selected for dissociation using the following criteria: peptide monoisotopic precursor determination, charge state between 3–9, intensity greater than 5e4, and a 30 s dynamic exclusion window. Each precursor that passed the selection criteria was subjected to subsequent HCD and ETD MS2 scans (resolution: 30000, quadrupole isolation window: 1.6 m/z units, HCD NCE: 28%, HCD AGC Target: 1.0e5, HCD max injection time: 150 ms, ETD collision time: calibrated charge dependent ETD parameters, ETD supplemental activation: 10% EThcD, ETD AGC Target: 2.0e5, ETD max injection time: 200 ms). Nine product ion scans of each type were performed for each precursor scan.

Separate peaklists were generated for ETD and HCD scans using Proteome Discoverer 2.2 (Thermo) and searched using Protein Prospector 5.23.0 (*66*). The search database consisted of the sequences of 82 human ribosomal protein components in addition to 10 sequences corresponding to the membrane associated components: TMCO1, Nicalin, NOMO1, NOMO2, NOMO3, Sec61A1, Sec61A2, Sec61β, Sec61γ, TMEM147 and CCDC47. The sequence of TMCO1 contained the N-terminal 3xFlag tag (reported crosslinked residue numbers reference the endogenous sequence). The three NOMO isoforms are highly homologous and in most cases cross-links to NOMO could not be assigned a specific isoform. These proteins were confirmed to be the dominant components of the sample by MS analysis of late eluting SEC fractions (corresponding to linear peptides). The 92 target proteins were concatenated with a decoy database consisting of 10 randomized amino acid sequences of for each target sequence (1012 total protein sequences searched). ETD peaklists were searched using Prospector instrument type ESI-ETD-high-res and HCD peaklists were searched using ESI-Q-high-res. Other search parameters were: mass tolerance of 7 ppm (precursor) and 15 ppm (product); fixed modifications of carbamidomethylation on cysteine; variable modifications of peptide N-terminal glutamine conversion to pyroglutamate, oxidation of methionine, and 'dead-end' modification of lysine and the protein N-terminus by semi-hydrolyzed BS3, protein N-terminal acetylation, protein N-terminal methionine loss, and incorrect monoisotopic precursor selection (neutral

loss of 1 Da); crosslinking reagent was DSS/BS3; trypsin specificity was used with three missed clea-vages and three variable modifications per peptide were allowed. The top 85 product ion signals were used for the search. Searches were performed using 64 cores on an HPC cluster and took about 4 hr to complete.

Cross-link spectral matches (CSM) were initially kept with peptide scores above 20, score differ-ence above 0, and length of each peptide between 4–25 residues. A linear support vector machine (SVM) model was constructed to classify CSMs between decoy and target classes (*Trnka et al., 2014*). Features selected for the SVM classifier were: score difference, percent of ions matched, pre-cursor charge state, rank of each peptide, and length of each peptide. Models were trained on half of the dataset and parameters were chosen to give a specificity of 90% tested on the other half of the data. Separate classifiers were built for ETD and HCD results. The best scoring CSM per unique cross-linked residue pair was selected and the ETD and HCD results were merged. The distribution of cross-linked residue pairs with one and two incorrectly identified peptides was modeled using essentially the same logic as (*Fischer and Rappsilber, 2017*), but extending their analysis to account for the 10x increased size of decoy database. The number of target-target hits with one wrong pep-tide is given by:

$$tf(TT) = (1/k)^*TD - (2/k^2)^*DD$$

and the number with both wrong is given by:

$$ff(TT) = (1/k^2)^*DD$$

where TT, TD, and DD are the number of target-target, target-decoy, and decoy-decoy hits, and k is the scaling factor describing the ratio in size of the decoy database to size of the target data-base. In this case k = 10. The final list of cross-links was reported at an SVM score of 1.5 which corre-sponded to a 0.55% FDR. Distance analysis was performed by measuring the Cα-Cα distances between all ribosome cross-links against an EM reconstruction of the human 80S ribosome (PDB ID 4ug0). At the reporting threshold of 1.5, the violation rate (fraction of mappable cross-links > 35 Å) was 8.7%.

## Cryo-EM sample preparation and data acquisition

Quantifoil 1.2/1.3 200 mesh grids coated in a 2 nm carbon film (Ted Pella, Inc) were glow discharged for 30 s immediately before use. Using an FEI Vitrobot, 2.5 μl of ~100 nM sample was applied to each grid, which was then incubated for 15 s at 22℃ and 100% humidity, blotted for 11 s, and flash frozen in liquid ethane. Data were collected on an FEI Titan Krios at 300 KV using Latitude S (Gatan) software, targeting defocus values from −2.5 to −1.0 μm. Exposure movies were recorded using a Gatan K3 energy filter and direct electron detector in super resolution mode at 64,000x magnifica-tion (super resolution pixel size of 0.68 Å) and a total exposure of 50e⁻/Å² fractionated over 40 frames.

## Data processing

5562 super resolution movies were summed and motion corrected using Motioncor2 (*Zheng et al., 2017*) with 2x binning, generating corrected micrographs with a pixel size of 1.36 Å. Contrast trans-fer function (CTF) parameters were estimated using GCTF (*Zhang, 2016*). 1,049,128 Particles were picked using the semi-autonomous particle picking algorithm in Relion3.1 (*70*). All 2D classification, 3D classification, and 3D refinement steps were performed in Relion3.1. Reference-free 2D classifica-tion was used to discard non-ribosome containing particles. An initial round of 3D classification using a reference 80S ribosome (EMD-5592) low pass filtered to 60 Å as an initial model was used to iso-late particles with clear ribosomal features. Particles in the best classes from this initial round of clas-sification were further examined via a second round of 3D classification against the same initial model. Five classes from this second round of 3D classification showed clear density for 40S and 60S ribosomal subunits, tRNAs, and luminal density below the micelle. Particles from these classes were used for an initial 3D refinement (286,091 particles). CTF refinement was used to estimate beamtilt across the dataset and refine per-particle defocus values. The ribosome density in this initial map was further refined by focused refinement using local angular searches and a mask around the ribo-some density (*Figure 2—figure supplement 3*, yellow mask). Non-translocon density was then

removed from the particle set using signal subtraction and a mask surrounding the translocon (*Figure 2—figure supplement 3*, cyan mask). These signal subtracted particles were subjected to 3D classification without alignment. One of the classes from this classification showed strong density for the TMCO1 translocon (82,684 particles, 28.8%). Particles from this class were refined using a translocon mask and local angular searches, producing a 3.8 Å reconstruction (Map 1, EMD-21426). We also reverted these particles to their original, ribosome-containing state for further analysis without signal subtraction. This produced a 3.4 Å reconstruction of the entire ribosome-translocon complex (Map 2, EMD-21427). Focused refinement with a translocon mask (*Figure 2—figure supplement 3*, magenta mask) and local angular searches produced a 3.8 Å reconstruction with improved translocon density (Map 3, EMD-21435). Where noted, maps were sharpened by applying a B-factor determined by the automated methods implemented in Relion3.1 (*70*). Additionally, local resolution estimation and filtering was performed using automated methods implemented in Relion3.1.

## Model building and refinement

We used the 60S ribosomal subunit, A/P and P/E tRNAs and the nascent chain from the human 80S ribosome-nascent chain complex structure (PDB ID 6OM0), and the Sec61 complex from the mammalian 80S ribosome-Sec61-OST structure (PDB ID 6FTI) as starting points for model building. Homology models for TMEM147 and Nicalin were generated in iTasser (*Yang et al., 2015*), using the γ-secretase subunits, APH-1 and Nicastrin, as templates (PDB ID 5A63). TMCO1 and CCDC47 models were generated with RaptorX-Contact (*Wang et al., 2017*; *Xu, 2019*).

All three maps were used for model building. The 60S ribosomal subunit (with tRNAs and poly-Ala nascent chain) was initially fitted as a rigid body into the 3.4 Å globally refined map (Map 2) (sharpened and low-pass filtered by local resolution) using UCSF Chimera (*Pettersen et al., 2004*), and then manually adjusted with rigid-body and real-space refinement using COOT 0.9-pre (*Emsley et al., 2010*). The Sec61 and TMEM147-Nicalin complexes, TMCO1 and CCDC47 were placed into focused (Map 3) and signal subtracted (Map 1) maps (unsharpened, and low-pass filtered by local resolution) as rigid bodies, and then adjusted using tightly restrained real-space refinement in COOT. No density was assigned to NOMO or the CCDC47 lumenal and transmembrane domains.

Real-space refinement of the model (60S and translocon) was done with PHENIX (*Afonine et al., 2018*), against the focused map (Map 3) (unsharpened, and low-pass filtered by local resolution). Three rounds of global minimization and group B-factor refinement were performed with tight secondary structure, reference model, rotamer, and Ramachandran restraints applied. Secondary structure- and reference model restraints were determined from the starting models. Hydrogen-bonding and base-pair and stacking parallelity restraints were applied to the rRNA. Final model statistics are provided in *Table 1*. Structure figures were generated with UCSF Chimera and PyMOL (http://www.pymol.org).

## RIP-seq analysis

Affinity purified TMCO1-ribosome complexes were isolated as described above ('Isolation of TMCO1-ribosome complexes for interaction analysis'), with the following changes. ~ $10^8$ cells were processed for each of three biological replicates. All buffers were made using DEPC-treated RNase free water. Solubilized membranes were incubated with M2 Flag affinity gel (Sigma, A2220) for 2 hr at 4°C with end-over-end mixing. To remove contaminating DNA, 1 U RNase-Free DNase (Promega, M6101) was added to the sample during resin binding. Unbound material was removed and the resin was washed four times with five column volumes of wash buffer to remove contaminating ribosomes. TMCO1-ribosome complexes were isolated by centrifugation as before. After centrifugation, the final pellet was resuspended in 250 mM sucrose, 300 mM potassium acetate, 50 mM Hepes pH 7.4, 5 mM magnesium acetate, 50 µg/mL emetine and 0.1% digitonin, flash frozen and stored at −80°C until ready for sequencing.

All mRNA sequencing was performed at the University of Chicago Genomics Facility. For each of three biological replicates, RNA was extracted, ribosomal RNA was removed by RiboZero and cDNA libraries were prepared. Fragment sizes were determined by Bioanalyzer, and samples were pooled for sequencing on an Illumina HiSeq 4000. 50 bp single-end sequence reads were aligned to the human GRCh38 reference transcriptome using STAR (v2.6.1) (*Dobin et al., 2013*) and gene transcript abundance was quantified by featureCounts using the Subread package (v1.6.3) (*Liao et al.,*

*2013*). Possible batch effects were adjusted using the SVA package (*Leek et al., 2012*) in R. An IP enrichment score was calculated as follows: IP enrichment = (Flag IP abundance - Control IP abundance)/Total membrane abundance, where 'Total membrane abundance' was determined by mRNA sequencing the total membrane fraction in HEK293 TRex cells, as described below. Only genes with mean CPM higher than 0.5 were considered confidently identified and used in the analysis.

For mRNA sequencing of total membrane-associated mRNAs, membrane suspensions from three biological replicates of parental HEK293 TRex cells were prepared as above, with the inclusion of 1 U/mL SuperaseIn and 50 µg/mL emetine at all times. Membranes were washed twice with 250 mM sucrose, 150 mM potassium acetate, 50 mM Hepes pH 7.4, 5 mM magnesium acetate, 50 µg/mL emetine, and then RNA was Trizol extracted. For each biological replicate, ribosomal RNA was removed by Oligo-dT affinity purification and cDNA libraries were prepared, sequenced, and analyzed as described above.

### Analysis of membrane protein expression levels and glycosylation patterns

For each replicate of the expression analysis, 750,000 cells were plated on poly-L-lysine coated plates and grown overnight. Cells were harvested by centrifugation, lysed using RIPA buffer (1% triton, 0.5% deoxycholate, 0.1% SDS and 1X protease inhibitor cocktail), and EAAT1 protein levels analyzed by SDS-PAGE and western blotting. Immunoblots were quantified using ImageJ (*Schneider et al., 2012*).

For EAAT1, Integrin α5 and TRAPα glycosylation analysis, RIPA cell lysates were reduced with 2% β−mercaptoethanol and denatured by heating for 10 min at 65˚C. Samples were then incubated with EndoH (NEB) or PNGaseF (Promega) for 7 hr at 37˚C, and analyzed by SDS-PAGE and western blotting.

For mRNA quantitation, total RNA was Trizol extracted (Ambion). cDNA (1000 ng) was synthesized using gDNA Clear cDNA Synthesis Kit (Bio-rad). qPCR was performed using iTaq Universal SYBR Green Supermix (Bio-rad) via CFX96 Touch Real-Time PCR Detection System (Bio-rad). Primers used for mRNA quantification were: EAAT1 fwd 5'-TTCCTGGGGAACTTCTGATG-3', EAAT1 rev 5'-CCATCTTCCCTGATGCCTTA-3', GAPDH fwd 5'-ACAACTTTGGTATCGTGGAAGG-3', and GAPDH rev 5'-GCCATCACGCCACAGTTTC-3'.

## Acknowledgements

We thank M Zhao, E Perozo, R Zhang, S Shao, B Reddy, T Edwards, U Baxa, A Weir, J Austin and T Lavoie for advice with EM data collection and analysis, A DiGuilio for cell lines, H Pourreza and staff at the UChicago Resource Computing Center for technical support, R Hegde for advice, reagents and comments on the manuscript, and Keenan lab members for discussions. This work was supported by NIH R21 EY026719 and NIH R01 GM130051 to RJK; SAA was supported by a Boehringer Ingelheim Fonds Ph.D. fellowship; PTM and FZ were supported by NIH training grant T32 GM007183. This research was also supported by the NCI National Cryo-EM Facility at the Frederick National Laboratory for Cancer Research under contract HSSN261200800001E, by the Dr. Miriam and Sheldon G Adelson Medical Research Foundation (AMRF), the NIH-NIGMS and UCSF Program for Breakthrough Biomedical Research (PBBR), which fund the UCSF Mass Spectrometry Facility, and by the NIH National Center for Advancing Translational Sciences through Grant Number 5UL1TR002389-02, which funds the UChicago Institute for Translational Medicine.

## Additional information

### Funding

| Funder | Grant reference number | Author |
| --- | --- | --- |
| National Institute of General Medical Sciences | NIH R01 GM130051 | Robert J Keenan |
| National Eye Institute | NIH R21 EY026719 | Robert J Keenan |
| Boehringer Ingelheim Fonds | PhD fellowship | S Andrei Anghel |

| National Institute of General Medical Sciences | T32 GM007183 | Philip T McGilvray Frank Zhong |
| Dr. Miriam and Sheldon G. Adelson Medical Research Foundation | | Alma L Burlingame |

The funders had no role in study design, data collection and interpretation, or the decision to submit the work for publication.

## Author contributions

Philip T McGilvray, Conceptualization, Data curation, Software, Formal analysis, Validation, Investigation, Visualization, Methodology, Writing - review and editing; S Andrei Anghel, Conceptualization, Data curation, Formal analysis, Funding acquisition, Validation, Investigation, Visualization, Methodology, Writing - review and editing; Arunkumar Sundaram, Formal analysis, Validation, Investigation, Methodology, Writing - review and editing; Frank Zhong, Investigation, Writing - review and editing; Michael J Trnka, Data curation, Software, Formal analysis, Validation, Investigation, Visualization, Methodology, Writing - review and editing; James R Fuller, Formal analysis, Writing - review and editing; Hong Hu, Data curation, Formal analysis, Writing - review and editing; Alma L Burlingame, Resources, Supervision, Funding acquisition, Writing - review and editing; Robert J Keenan, Conceptualization, Resources, Data curation, Formal analysis, Supervision, Funding acquisition, Validation, Investigation, Visualization, Writing - original draft, Project administration, Writing - review and editing

## Author ORCIDs

Philip T McGilvray (iD) https://orcid.org/0000-0002-2693-0076
Michael J Trnka (iD) http://orcid.org/0000-0002-8808-5146
James R Fuller (iD) http://orcid.org/0000-0002-9029-0923
Robert J Keenan (iD) https://orcid.org/0000-0003-1466-0889

## Decision letter and Author response

Decision letter https://doi.org/10.7554/eLife.56889.sa1
Author response https://doi.org/10.7554/eLife.56889.sa2

# Additional files

## Supplementary files

• Transparent reporting form

## Data availability

Annotated spectra corresponding to the reported ribosome-translocon cross-links are available at the MS-Viewer website (http://msviewer.ucsf.edu/prospector/cgi-bin/msform.cgi?form=msviewer) with the following accession keys: HCD data: 7s2yb4zfjw and ETD data: vdibnsypj7. Cryo-EM maps have been deposited in the Electron Microscopy Data Bank with accession codes: EMD-21426 (Map 1), EMD-21427 (Map 2) and EMD-21435 (Map 3). Coordinates for the human 60S-translocon complex have been deposited in the Protein Data Bank with accession code 6W6L. mRNA sequencing data have been deposited in Gene Expression Omnibus (GEO) under accession number GSE134027.

The following datasets were generated:

| Author(s) | Year | Dataset title | Dataset URL | Database and Identifier |
|---|---|---|---|---|
| McGilvray PT, Anghel SA, Sundaram A, Zhong F, Trnka MJ, Fuller JR, Hu H, Burlingame AL, Keenan RJ | 2020 | Single Particle Cryo-EM Structure of the Natively Isolated Sec61 complex, TMCO1, Nicalin, TMEM147, and CCDC47 Containing Translocon | https://www.ebi.ac.uk/pdbe/entry/emdb/EMD-21426 | Electron Microscopy Data Bank, EMD-21426 |

| McGilvray PT, Anghel SA, Sundaram A, Zhong F, Trnka MJ, Fuller JR, Hu H, Burlingame AL, Keenan RJ | 2020 | Single Particle Cryo-EM Structure of the Natively Isolated Sec61 complex, TMCO1, Nicalin, TMEM147, and CCDC47 Containing Translocon | https://www.ebi.ac.uk/pdbe/entry/emdb/EMD-21427 | Electron Microscopy Data Bank, EMD-21427 |
|---|---|---|---|---|
| McGilvray PT, Anghel SA, Sundaram A, Zhong F, Trnka MJ, Fuller JR, Hu H, Burlingame AL, Keenan RJ | 2020 | Single Particle Cryo-EM Structure of the Natively Isolated Sec61 complex, TMCO1, Nicalin, TMEM147, and CCDC47 Containing Translocon | https://www.ebi.ac.uk/pdbe/entry/emdb/EMD-21435 | Electron Microscopy Data Bank, EMD-21435 |
| McGilvray PT, Anghel SA, Sundaram A, Zhong F, Trnka MJ, Fuller JR, Hu H, Burlingame AL, Keenan RJ | 2020 | Cryo-EM structure of the human ribosome-TMCO1 translocon | https://www.rcsb.org/structure/6W6L | RCSB Protein Data Bank, 6W6L |
| McGilvray PT, Anghel SA, Sundaram A, Zhong F, Trnka MJ, Fuller JR, Hu H, Burlingame AL, Keenan RJ | 2020 | mRNA sequencing data | http://www.ncbi.nlm.nih.gov/geo/query/acc.cgi?acc=GSE134027 | NCBI Gene Expression Omnibus, GSE134027 |

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
