## [Decision Letter]

**Acceptance summary:**

Mechanistic explanations for polytopic transmembrane protein biogenesis remain incomplete. The textbook model of how the Sec61 translocon guides hydrophobic transmembrane segments from the ribosome into the ER membrane cannot yet account for how polytopic membrane proteins are inserted into the ER membrane with their proper topology, tertiary packing, or oligomerization status. In an exciting step towards closing this gap, McGilvray and Anghel et al. report the discovery of a massive, multi-component translocon-associated complex. This complex comprises TMCO1, Sec61α/β/γ, as well as CCDC47 and the Nicalin-TMEM147-NOMO complex. Together, this complex appears to harbor multiple routes into the ER membrane, and loss of the complex comprises the biosynthesis of polytopic membrane proteins.

**Decision letter after peer review:**

Thank you for submitting your article "An ER translocon for multi-pass membrane protein biogenesis" for consideration by *eLife*. Your article has been reviewed by three peer reviewers, including Adam Frost as the Reviewing Editor and Reviewer #1, and the evaluation has been overseen by Suzanne Pfeffer as the Senior Editor. The following individual involved in review of your submission has agreed to reveal their identity: Vladimir Denic (Reviewer #2).

The reviewers and senior editor have discussed the reviews with one another and the Reviewing Editor has drafted this decision to help you prepare a revised submission.

As the editors have judged that your manuscript is definitely of interest, but as described below that additional experiments are required before it is published, we would like to draw your attention to changes in our revision policy that we have made in response to COVID-19 (https://elifesciences.org/articles/57162). First, because many researchers have temporarily lost access to the labs, we will give authors as much time as they need to submit revised manuscripts. We are also offering, if you choose, to post the manuscript to bioRxiv (if it is not already there) along with this decision letter and a formal designation that the manuscript is 'in revision at *eLife*'. Please let us know if you would like to pursue this option. (If your work is more suitable for medRxiv, you will need to post the preprint yourself, as the mechanisms for us to do so are still in development.)

Summary:

Mechanistic explanations of polytopic membrane protein biogenesis remain incomplete. The textbook understanding of how Sec61 guides hydrophobic transmembrane segments from the ribosome into the ER membrane rests on detailed biochemistry and near-atomic resolution structures. The textbook model, however, cannot yet explain how polytopic membrane proteins are inserted into the ER membrane with their proper topology, tertiary transmembrane segment packing, or oligomerization because the Sec61 translocon can only accommodate, at most, two transmembrane helices. In an exciting step towards explaining the biogenesis of polytopic membrane proteins, McGilvray and Anghel et al. report the discovery of a new Sec61-associated complex with roles in polytopic membrane protein biogenesis.

Their story starts with biochemical characterization of proteins that copurify with detergent-solubilized TMCO1-a protein that the Keenan group has previously hypothesized to have a role in membrane protein biogenesis due to its homology with YidC/Oxa1 family proteins. Mass spectrometry and western blotting revealed that TMCO1 copurifies with 80S ribosomes, Sec61α/β/γ, as well as CCDC47 and the Nicalin-TMEM147-NOMO complex. Multi-class 3D reconstructions using cryoEM suggests an intricate assembly of these factors organized around the exit tunnel of the ribosome. Analysis of the transmembrane segments from the Sec61 complex, TMCO1, CCDC47, TMEM147, and Nicalin further suggests the complex is organized around a membrane cavity and that this organization may be consistent with multiple routes into the membrane, including re-entry from the lumenal face and additional pores flanking the canonical pore of the Sec61 complex. Finally, Riboseq measurements of the TMCO1-copurifying messages showed that the TMCO1-bound ribosomes are strongly enriched for multi-pass membrane proteins. Cells lacking components of this TMCO1 complex are unable to properly make a reporter client, the trimeric glutamate transporter EAAT1. In summary, the identification of a new translocation machine for multi-pass membrane protein biogenesis warrants publication by *eLife*. Several aspects of the author's structure/function model, however, are beyond the resolution limits of the current study and will require additional functional tests or less declarative claims.

Essential revisions:

1) The manuscript does not provide direct evidence that the novel Sec61-associated complex mediates the insertion of polytopic membrane proteins. The reduced levels of EAAT1 in cells lacking these factors are consistent with a role in membrane protein biogenesis, but reduced levels could be due by a defect in ER targeting and nascent protein aggregation/degradation. Among other approaches, the authors could assess proper translocation by monitoring the N-glycosylation of a nascent and topologically-defined substrate.

2) The mass spectrometry, western blots, and cryoEM maps may be consistent with the existence of two or more assemblies, rather than a single complex containing all five of the accessory proteins (TMCO1, CCDC47, Nicalin, TMEM147, and NOMO). Specifically, there are no cross-links between the TMEM147-Nicalin-NOMO complex and Sec61/CCDC47/TMCO1. Thus, while the cross-linking demonstrates that TMCO1 and CCDC47 engage the ribosome and Sec61, one simple explanation of the remaining hybrid peptides is that the TMEM147/Nicalin/NOMO complex is distinct from Sec61/CCDC47/TMCO1 complex. Without conclusive functional or biochemical characterization to demonstrate that this is indeed a unitary complex, this claim should be removed. Most importantly, providing biochemical evidence for the presence of TMEM147 in the TMCO1 complex is an essential request.

3) A recurrent and motivating theme of the manuscript concerns the claim that TMCO1 possesses a "three-TMD core, which harbors a groove lined with conserved charged and polar residues…the TMCO1 groove opens directly into the lipid-filled cavity at the center of the translocon, revealing a pathway by which a hydrophobic segment could be inserted from the cytosol into a protected membrane environment." This notion is attractive speculation, but at this resolution, the structural evidence is not sufficiently strong to support the model. While Figure 2E suggests a groove, at this resolution we cannot place the charged or polar residues with confidence. The RaptorX-based modeling, as summarized in Figure 2—figure supplement 7, appears compelling, but both the surface properties and roles of this groove remain unresolved. Without testing loss-of-function mutants along the groove, this aspect of the structural model is unsupported. Considering that the architecture of the bacterial holotranslocon is strikingly different in terms of the relationship between the YidC groove and SecYEG, show in Figure 4—figure supplement 1, it is premature to conclude that TMCO1 functions as proposed.

4) Related to point 3, the transmembrane region appears to be ~7Å on average in resolution. The cross-links determined by mass spectrometry are helpful constraints, but from Figure 2 and its associated supplements we cannot be sure the TMs have all been placed correctly. Among other uncertainties, for example, the NOMO C-term has extensive cross-links with Sec61γ. Could NOMO be at the center of the assembly, near Sec61γ, rather than CCDC47? Similarly, how reliable is the assignment of the Nicalin TM versus the TMs of TMEM147? At this resolution, side chains and many loops are invisible or, at best, ambiguous.

[Editors' note: further revisions were suggested prior to acceptance, as described below.]

Thank you for submitting your article "An ER translocon for multi-pass membrane protein biogenesis" for consideration by *eLife*. Your article has been re-reviewed by the original peer reviewers, and the evaluation has been overseen by Reviewing Editor Adam Frost and Suzanne Pfeffer as the Senior Editor.

The reviewers have discussed the reviews with one another and the Reviewing Editor has drafted this decision to help you prepare a revised submission.

We would like to draw your attention to changes in our revision policy that we have made in response to COVID-19 (https://elifesciences.org/articles/57162). Specifically, when editors judge that a submitted work as a whole belongs in *eLife* but that some conclusions require a modest amount of additional new data, as they do with your paper, we are asking that the manuscript be revised to either limit claims to those supported by data in hand, or to explicitly state that the relevant conclusions require additional supporting data. In this case, the reviewers felt that one issue could be dealt with textually but they required one additional experiment be carried out prior to formal acceptance. We hope you will find these comments constructive.

We are pleased to accept your work for publication, pending the resolution of two issues.

First, the revised manuscript does not directly demonstrate that the reduced expression of a putative client, EAAT1, in TMCO1 KO cells is due to a membrane insertion defect. It remains possible, if unlikely, that clients like EAAT1 depend on the TMCO1 complex for ER targeting rather than membrane insertion. Your new experiments show that loss of TMCO1 does not lead to a general deficiency in ER targeting for clients that are not expected to depend on the TMCO1-containing complex (single-pass membrane proteins); but falls short of demonstrating client insertion via the TMCO1, CCDC47, Nicalin, TMEM147 and NOMO complex. We ask you to acknowledge this caveat in your Discussion section, perhaps following the sentence, "The biochemical function of the TMCO1 translocon is unclear," and before the sentence that begins, "The simplest possibility…".

Second, the absence of SDS-PAGE, protein blot or mass spectrometry data indicating that TMEM147 co-purifies with the holocomplex belies the claim that "…a cluster of eight TMDs visible near the Sec61 hinge were unambiguously assigned to the Nicalin-TMEM147 sub-complex". The placement of the TMEM147-Nicalin helices is such a central aspect of the structural model that given the resolution limits in the membrane, independent biochemical evidence for the presence of TMEM147 within the complex remains an essential request, despite the informative hybrid peptides between TMEM147 and the ribosome. We note that there are literature reports of ER-localized, N-terminal epitope-tagged TMEM147 reagents that may be useful in this regard.

---

## [Author Response]

Essential revisions:1) The manuscript does not provide direct evidence that the novel Sec61-associated complex mediates the insertion of polytopic membrane proteins. The reduced levels of EAAT1 in cells lacking these factors are consistent with a role in membrane protein biogenesis, but reduced levels could be due by a defect in ER targeting and nascent protein aggregation/degradation. Among other approaches, the authors could assess proper translocation by monitoring the N-glycosylation of a nascent and topologically-defined substrate.

To address the possibility that reduced EAAT1 levels in cells lacking TMCO1 components is due to a general defect in ER targeting, insertion or glycosylation, we monitored the biogenesis of two single-pass membrane proteins, integrin α5 and TRAPα, in wild-type and TMCO1 knockout cells. These secretory pathway transmembrane proteins are not expected to be clients of the TMCO1 translocon. As shown in a modified Figure 5—figure supplement 1, we see no change in the steady-state expression levels of either protein. Moreover, analysis of EndoH and PNGase sensitivity is consistent with proper trafficking in the TMCO1 knockout cells. While the precise biochemical function(s) of the TMCO1 translocon remains to be defined, these data indicate that disruption of TMCO1 does not lead to a general defect in membrane protein biogenesis.

2) The mass spectrometry, western blots, and cryoEM maps may be consistent with the existence of two or more assemblies, rather than a single complex containing all five of the accessory proteins (TMCO1, CCDC47, Nicalin, TMEM147, and NOMO). Specifically, there are no cross-links between the TMEM147-Nicalin-NOMO complex and Sec61/CCDC47/TMCO1. Thus, while the cross-linking demonstrates that TMCO1 and CCDC47 engage the ribosome and Sec61, one simple explanation of the remaining hybrid peptides is that the TMEM147/Nicalin/NOMO complex is distinct from Sec61/CCDC47/TMCO1 complex. Without conclusive functional or biochemical characterization to demonstrate that this is indeed a unitary complex, this claim should be removed. Most importantly, providing biochemical evidence for the presence of TMEM147 in the TMCO1 complex is an essential request.

Our previous attempts to identify ribosomes engaged with translocon sub-complexes (e.g., Sec61-CCDC47-TMCO1 or Sec61-Nicalin-TMEM147-NOMO) in the natively-isolated samples by cryoEM were unsuccessful, despite extensive efforts at further *in silico* classification. Nevertheless, to more rigorously show that these ribosome-associated accessory components are part of a unitary complex, we repeated the original purifications using cells overexpressing stably integrated 3xFlag-TMCO1 or 3xFlag-Nicalin. As shown in a new Figure 1—figure supplement 1, TMCO1, CCDC47 and NOMO all co-purify in the ribosome-bound fraction following 3xFlag-Nicalin immunoprecipitation. This provides further evidence that these components can be isolated as a single, ribosome-associated complex. We also show in a new Figure 1—figure supplement 1 that in the absence of ribosomes, only components of the known Nicalin-TMEM147-NOMO complex remain associated, indicating that TMCO1, CCDC47 and the pre-formed Nicalin-TMEM147-NOMO assemble in the context of the ribosome.

None of the commercial anti-TMEM147 antibodies we tested could detect TMEM147, whether in total lysate, microsomal membrane preparations, or natively isolated TMCO1-ribosome complexes. This is likely because the longest extra-membrane TMEM147 loop is only 5 residues long, making it difficult to detect by western blot. Nevertheless, several lines of evidence support the conclusion that TMEM147 is part of the ribosome-bound TMCO1 translocon: (1) previous studies demonstrate that TMEM147 is a core component of the Nicalin-NOMO complex (Dettmer et al., 2010); (2) we observe a cluster of crosslinked peptides between TMEM147 and uL24 by mass spectrometry (between K94 of TMEM147 and K110, K113 and K116 of uL24); and (3) we visualize all seven predicted TMEM147 TMs in the cryoEM maps (additional view now shown in an expanded Figure 2—figure supplement 5). Even at moderate resolution these helical densities are observed to form an unusual inverted-V shape. TMEM147 and Nicalin are evolutionarily related to the APH1-Nicastrin subcomplex of human γ-secretase, for which high resolution structures are available (e.g., PDB ID 5A63), and in these structures, the seven TMs of APH1 adopt a similarly distinctive, inverted-V shape. Moreover, assigning TMEM147 to this density places the single Nicalin TM into helical density at precisely its evolutionarily predicted location – packed against TM1 of TMEM147. Thus, even in the absence of additional biochemical evidence, our data strongly suggest that TMEM147 is a bona fide component of the TMCO1 translocon. We clarify these points in the main text.

3) A recurrent and motivating theme of the manuscript concerns the claim that TMCO1 possesses a "three-TMD core, which harbors a groove lined with conserved charged and polar residues…the TMCO1 groove opens directly into the lipid-filled cavity at the center of the translocon, revealing a pathway by which a hydrophobic segment could be inserted from the cytosol into a protected membrane environment." This notion is attractive speculation, but at this resolution, the structural evidence is not sufficiently strong to support the model. While Figure 2E suggests a groove, at this resolution we cannot place the charged or polar residues with confidence. The RaptorX-based modeling, as summarized in Figure 2—figure supplement 7, appears compelling, but both the surface properties and roles of this groove remain unresolved. Without testing loss-of-function mutants along the groove, this aspect of the structural model is unsupported. Considering that the architecture of the bacterial holotranslocon is strikingly different in terms of the relationship between the YidC groove and SecYEG, show in Figure 4—figure supplement 1, it is premature to conclude that TMCO1 functions as proposed.

We have revised the manuscript to temper statements that are not directly supported by the data. This includes removing references to the hydrophilic/charged funnel in TMCO1, and explicitly stating in the Discussion that “[t]he biochemical function of the TMCO1 translocon is unclear.” The key points here are: (i) we have shown that TMCO1 is structurally similar to other members of the Oxa1 superfamily of membrane biogenesis factors; (ii) that it possesses a cytosolic funnel that opens to the membrane; and (iii) that it lines a large, lipid-filled cavity at the heart of the translocon. These points are unambiguous, even in the absence of TMCO1 side chain density, and they lead us to propose that the TMCO1 translocon functions as an insertase and an intramembrane chaperone. Future mechanistic studies will test this model.

That YidC and TMCO1 might assemble in different orientations relative to the Sec complex in these two very different assemblies is not a contradiction. On the contrary: members of the superfamily are highly divergent (~10-20% sequence identity) and have evolved to function in very different contexts. Indeed, YidC is thought to function co- and post-translationally, in both Sec-dependent and Sec-independent modes. EMC3 functions as part of a large multi-subunit complex in both co- and post-translational modes yet doesn’t appear to directly contact Sec61 or ribosomes. And Get1 functions as part of a dedicated post-translational insertase in complex with Get2. Despite these different contexts, a core function of these proteins – moving transmembrane segments into the bilayer – appears to be conserved. We now clarify this point in the Discussion.

Nevertheless, we also opted to remove the bacterial holotranslocon comparison from the text. This is for two reasons. First, this model is derived from a low (~14 Å overall) resolution cryoEM reconstruction, making it difficult to reliably assign the six different bacterial subunits to the density. Second, the bacterial holotranslocon structure was obtained in the absence of the ribosome, which may alter the relative arrangement of YidC and SecYEG. Although there are no structures of a ribosome-SecYEG-YidC complex, we note that a low resolution cryoEM structure of YidC (by itself) bound to a ribosome-nascent chain complex suggests that YidC contacts L24 (Kedrov et al., 2016), which is consistent with the location of TMCO1 in our structure (bacterial L24 ~ human uL24). Thus, we decided to remove the apples-to-oranges comparison with the bacterial holotranslocon.

4) Related to point 3, the transmembrane region appears to be ~7Å on average in resolution. The cross-links determined by mass spectrometry are helpful constraints, but from Figure 2 and its associated supplements we cannot be sure the TMs have all been placed correctly. Among other uncertainties, for example, the NOMO C-term has extensive cross-links with Sec61γ. Could NOMO be at the center of the assembly, near Sec61γ, rather than CCDC47? Similarly, how reliable is the assignment of the Nicalin TM versus the TMs of TMEM147? At this resolution, side chains and many loops are invisible or, at best, ambiguous.

We are able to unambiguously place TMCO1, CCDC47, Nicalin and TMEM147 into the moderate resolution translocon density using deep-learning based structure prediction methods (TMCO1 and CCDC47), homology modeling based on the previously described evolutionary relationship between TMEM147-Nicalin and the APH1-Nicastrin subunits of γsecretase (Dettmer et al., 2010), and the distinctive features of the resulting models. These assignments are fully consistent with the XL-MS, which was not used to provide additional constraints during the modeling, but to validate the final model.

The main ambiguity in our model lies with the single-pass NOMO subunit. Weak density is visible adjacent to TM2 and TM6 of TMEM147, where the unstructured ~46 residue cytosolic C-terminus of NOMO would be well-positioned to satisfy the experimentally observed crosslinks with the ribosomal subunits eL19, eL24, eL38, uL29, uL23 and the N-terminal region of Sec61γ. In the absence of additional density, however, we chose not to assign NOMO.

Regardless of the exact location of NOMO, the cytosolic domain of CCDC47 is unambiguously assigned, as evidenced by the fit of its distinctive coiled-coil and globular domains (the cytosolic region of NOMO is too small), and a series of regio-specific crosslinks to ribosomal subunits, TMCO1 and Sec61. Similarly, placement of the seven TMs of TMEM147 and the single Nicalin TM into density is unambiguous (see Response #2), satisfying the evolutionary constraints provided by homology to γ-secretase, and positioning the large lumenal domain of Nicalin into the low-resolution lobe of density on the lumenal side of the translocon. This assignment is also consistent with the experimentally observed cross-links between TMEM147 and uL24.

We clarify these points throughout the manuscript, including an expanded Figure 2—figure supplement 5, that provides more detail for the map-to-model fits.

[Editors' note: further revisions were suggested prior to acceptance, as described below.]

We are pleased to accept your work for publication, pending the resolution of two issues.First, the revised manuscript does not directly demonstrate that the reduced expression of a putative client, EAAT1, in TMCO1 KO cells is due to a membrane insertion defect. It remains possible, if unlikely, that clients like EAAT1 depend on the TMCO1 complex for ER targeting rather than membrane insertion. Your new experiments show that loss of TMCO1 does not lead to a general deficiency in ER targeting for clients that are not expected to depend on the TMCO1-containing complex (single-pass membrane proteins); but falls short of demonstrating client insertion via the TMCO1, CCDC47, Nicalin, TMEM147 and NOMO complex. We ask you to acknowledge this caveat in your Discussion section, perhaps following the sentence, "The biochemical function of the TMCO1 translocon is unclear," and before the sentence that begins, "The simplest possibility…".

We have modified the text accordingly.

Second, the absence of SDS-PAGE, protein blot or mass spectrometry data indicating that TMEM147 co-purifies with the holocomplex belies the claim that "…a cluster of eight TMDs visible near the Sec61 hinge were unambiguously assigned to the Nicalin-TMEM147 sub-complex". The placement of the TMEM147-Nicalin helices is such a central aspect of the structural model that given the resolution limits in the membrane, independent biochemical evidence for the presence of TMEM147 within the complex remains an essential request, despite the informative hybrid peptides between TMEM147 and the ribosome. We note that there are literature reports of ER-localized, N-terminal epitope-tagged TMEM147 reagents that may be useful in this regard.

Our initial attempts to immunoprecipitate the ribosome-associated assembly via an N-terminally 3xFlag-tagged TMEM147 were unsuccessful – possibly because the N-terminal tag is inaccessible to the antibody in the context of the TMCO1 translocon, or because the tag prevents assembly of the intact ribosome-translocon complex. However, after screening additional commercially available antibodies, we succeeded in finding one that specifically recognizes endogenous TMEM147 (confirmed by blotting against recombinant 3xFlag-tagged TMEM147). As shown now in a modified Figure 1C, TMEM147 co-purifies with the TMCO1 translocon.